# DIRAS1 Drives Oxaliplatin Resistance in Colorectal Cancer via PHB1-Mediated Mitochondrial Homeostasis

**DOI:** 10.3390/biology14070819

**Published:** 2025-07-05

**Authors:** Min Long, Qian Ouyang, Jingyi Wen, Xuan Zeng, Zihui Xu, Shangwei Zhong, Changhao Huang, Jun-Li Luo

**Affiliations:** 1The Cancer Research Institute, Hengyang Medical School, University of South China (USC), Hengyang 421001, China; 13677484460@163.com (M.L.); o17352698397@163.com (Q.O.); hai17872762620@163.com (X.Z.); xuzihui0022@126.com (Z.X.); swzhong@usc.edu.cn (S.Z.); 2MOE Key Laboratory of Rare Pediatric Diseases, Hengyang Medical School, University of South China (USC), Hengyang 421001, China; 3Department of Organ Transplantation, Xiangya Hospital, Central South University, Changsha 410008, China; 4National Health Commission Key Laboratory of Birth Defect Research and Prevention, Hunan Provincial Maternal and Child Health Care Hospital, University of South China (USC), Changsha 410008, China; 5Hunan Provincial Key Laboratory of Basic and Clinical Pharmacological Research of Gastrointestinal Cancer, University of South China (USC), Hengyang 421001, China

**Keywords:** colorectal cancer, oxaliplatin resistance, DIRAS1, PHB1, mitochondrial function, targeted therapy

## Abstract

Oxaliplatin (OXA) resistance remains a major challenge in colorectal cancer (CRC) chemotherapy, with approximately 15 to 50% of stage III patients developing resistance to this frontline drug. This study explores the role of DIRAS1, a RAS family protein with previously undefined relevance in CRC, in mediating OXA resistance mechanisms. Through a combination of in vitro assays (MTT, wound healing, colony formation), transcriptomic profiling, and in vivo mouse models, we demonstrate that DIRAS1 expression is elevated following prolonged chemotherapy and is positively correlated with OXA resistance. Silencing DIRAS1 reduced OXA IC_50_ and enhanced tumor sensitivity both in vitro and in vivo. Mechanistically, DIRAS1 promotes chemoresistance by upregulating PHB1, which stabilizes mitochondrial function. Clinical specimen analysis further validated the clinical relevance of the DIRAS1–PHB1 pathway. These findings suggest that targeting the DIRAS1–PHB1 axis may represent a promising strategy to overcome OXA resistance, reduce recurrence, and improve outcomes for CRC patients.

## 1. Introduction

Colorectal cancer (CRC) is one of the most prevalent malignancies of the digestive system worldwide, with a steadily rising incidence and mortality rate. According to data from the International Agency for Research on Cancer (IARC), CRC ranked third in incidence and second in mortality among all cancers globally in 2022, posing a substantial threat to public health and life expectancy [1]. Current standard treatment primarily involves surgical resection, supplemented by chemotherapy and radiotherapy [2]. Oxaliplatin (OXA)-based regimen is a first-line chemotherapy by inducing DNA crosslinking [3] and triggering mitochondrial-mediated apoptosis [4,5,6]. However, approximately 15% to 50% of stage III CRC patients experience postoperative recurrence, often associated with the development of OXA resistance, severely compromising therapeutic efficacy [7]. Several mechanisms have been implicated in CRC chemoresistance, including enhanced drug efflux [8,9], detoxification by glutathione-S-transferases [10], increased DNA repair [8,11], and activation of anti-apoptotic pathways such as Bcl-2 upregulation [12]. Despite advances in uncovering these mechanisms, clinical outcomes for CRC patients with OXA resistance have shown limited improvement. Therefore, deeper mechanistic insights and the identification of novel therapeutic targets to overcome resistance are urgently needed.

The RAS superfamily of small GTPases plays essential roles in regulating cellular processes, including proliferation, differentiation, intracellular trafficking, and vesicular dynamics [13]. DIRAS1, a member of this family, is a GTPase encoded by a gene located on chromosome 19p13.3, consisting of two exons and producing a 198-amino acid protein [14,15]. DIRAS1 expression is frequently downregulated in tumors due to copy number loss, loss of heterozygosity, or promoter hypermethylation [16,17,18]. Although it has been described as a tumor suppressor in glioblastoma [14], CRC [19], renal cell carcinoma [20,21], and ovarian cancer [16], its prognostic and functional roles remain ambiguous [14,15].

Mitochondria are central to cellular metabolism and apoptosis, with mitochondrial membrane integrity playing a key role in drug sensitivity [22]. In CRC, mitochondrial dysfunction has been increasingly linked to chemoresistance. Altered mitochondrial membrane potential and permeability can impede apoptosis induction and contribute to reduced drug accumulation [5,23,24]. This alteration may aid cancer cells in adapting to chemotherapeutic stress. Targeting mitochondrial function has thus emerged as a potential strategy to enhance drug sensitivity [25,26].

Prohibitin 1 (PHB1) is a highly conserved protein localized in the inner mitochondrial membrane, where it regulates mitochondrial integrity and dynamics by interacting with proteins and lipids [27]. In intestinal epithelial cells, PHB1 deficiency leads to mitochondrial dysfunction and has been linked to inflammatory diseases such as Crohn’s disease [28,29]. Loss of PHB1 destabilizes mitochondrial structure, promotes mitochondrial DNA (mtDNA) release, and triggers immune activation [30]. It has been reported that PHB1 contributes to cancer drug resistance by regulating mitochondrial fission and fusion via Oma1 protease-mediated pathways [31], highlighting its pivotal role in maintaining mitochondrial homeostasis and cellular survival under stress.

In this study, we identify DIRAS1 as a key contributor to OXA resistance in CRC by modulating PHB1 expression and mitochondrial function. Targeting the DIRAS1–PHB1 axis may offer a novel therapeutic strategy to overcome chemoresistance in CRC.

## 2. Materials and Methods

### 2.1. Clinical Samples

A total of 50 patients diagnosed with CRC at Central South University between January 2019 and August 2024 were enrolled in this study. Paired tumor tissues and adjacent non-tumorous tissues were collected during surgical resection. Inclusion criteria were (i) histopathological confirmation of CRC, (ii) no prior oncologic treatment before surgery, and (iii) availability of complete clinical data. Post-treatment follow-up was performed, and relevant clinical characteristics are summarized in Appendix A.

To investigate mechanisms underlying OXA resistance, a subgroup of patients who received OXA-based adjuvant chemotherapy was carefully selected. Based on clinical response, these patients were categorized into a drug-sensitive group (*n* = 5) and a drug-resistant group (*n* = 5). The study protocol and the use of human CRC tissue samples were approved by the Research Ethics Committee of Xiangya Hospital, Central South University, and conducted in accordance with all applicable ethical guidelines. Informed consent was obtained from all participants prior to sample collection.

### 2.2. Cell Culture

Human cell lines 293T, HCT116, DLD1, and SW620 were obtained from Procell Life Science & Technology (Wuhan, China). Short tandem repeat (STR) profiling confirmed the identity and mycoplasma-free status of all cell lines. Cells were cultured in low-glucose Dulbecco’s Modified Eagle Medium (DMEM; Biological Industries, Beit Haemek, Israel) supplemented with 10% fetal bovine serum (FBS) and 100 μg/mL penicillin–streptomycin. Cultures were maintained in a humidified incubator at 37 °C with 5% CO_2_.

### 2.3. Cell Proliferation Assay

Cell proliferation was assessed using the MTT assay (Beyotime Biotechnology, Shanghai, China). Cells were seeded into 96-well plates at a density of 2 × 10^3^ cells/well. At 24, 72, and 120 h post-seeding, 10 μL of MTT reagent was added to each well and incubated for 4 h at 37 °C. Subsequently, 200 μL of DMSO was added to dissolve the formazan crystals, and absorbance was measured at 570 nm using a Multiskan FC microplate reader (Thermo Fisher Scientific, Shanghai, China). All experiments were conducted in triplicate.

### 2.4. Wound Healing Assay

To evaluate cell migration, a wound-healing assay was performed. Cells were plated in 12-well plates and grown to 100% confluence. A sterile 200 μL pipette tip was used to create a linear scratch in the monolayer. After removing cell debris by washing with PBS, plates were incubated under standard conditions. Images were acquired at 0 and 24 h post-scratch using a phase-contrast microscope [32].

### 2.5. Plasmid Constructs and Lentiviral Packaging

Plasmids for overexpression and knockdown of DIRAS1 and PHB1 were constructed and packaged into lentiviral particles. Full-length coding sequences (CDSs) of DIRAS1 and PHB1 were amplified and cloned into the LVCV-19 vector (Sino Biological Inc., Beijing, China) [33]. For gene knockdown, shRNA sequences targeting DIRAS1 and PHB1 were inserted into the pLKO.1 vector (Addgene, Watertown, MA, USA). Primer sequences are listed in Table 1. Lentiviral packaging was performed in 293T cells using second-generation packaging plasmids psPAX2 and pMD2.G (Addgene) through co-transfection [34]. Viral supernatants were collected and used to transduce target CRC cell lines for subsequent assays.

### 2.6. RNA Isolation and Semi-Quantitative Reverse Transcription PCR

Total RNA was extracted from CRC tissues using a low-temperature fragmentation technique followed by Trizol reagent (Takara, Kusatsu, Japan), according to the manufacturer’s protocol. Complementary DNA (cDNA) was synthesized using 1 µg of total RNA with the PrimeScript™ RT Master Mix (Takara, Kusatsu, Japan) following the recommended procedure.

### 2.7. Western Blotting (WB)

Cells were lysed using RIPA buffer (10 mM Tris-Cl pH 8.0, 150 mM NaCl, 1% Triton X-100, 1% sodium deoxycholate, 1 mM EDTA, 0.05% SDS) supplemented with protease inhibitor cocktail. Protein concentrations were quantified via the Bradford method (Bio-Rad, Hercules, CA, USA). Equal amounts of protein were separated by SDS-PAGE and transferred to Hybond ECL membranes (GE Healthcare, Chicago, IL, USA). Membranes were incubated overnight with primary antibodies against DIRAS1 (H681706012, Huabio, Hangzhou, China), PHB1 (db11502, Diagbio, Hangzhou, China), and GAPDH (sc-47484, Santa Cruz), followed by HRP-conjugated secondary antibodies. Signals were visualized using ECL or SuperSignal™ West Dura chemiluminescent substrates (Thermo Scientific, Waltham, MA, USA). All experiments were repeated in triplicate.

### 2.8. Immunohistochemistry (IHC)

Paraffin-embedded tissue sections (4 μm thick) were subjected to antigen retrieval and incubated overnight at 4 °C with anti-DIRAS1 antibody (1:200 dilution in 10% blocking serum). Following incubation with an HRP-conjugated secondary antibody (1:500) for 1 h at room temperature, DAB (Sigma, D12384, St. Louis, MO, USA) was used for signal detection, and nuclei were counterstained with Harris’ hematoxylin. IHC scores were determined by multiplying staining intensity (0 = negative, 1 = weak, 2 = moderate, 3 = strong) and percentage of positive cells (0 = 0–5%, 1 = 5–25%, 2 = 26–50%, 3 = 51–75%, 4 = > 75%) [35].

### 2.9. Annexin V-FITC/PI Apoptosis Assay

Cells were harvested, washed twice with cold PBS, and resuspended in binding buffer. Annexin V-FITC and propidium iodide (PI) were added, and samples were incubated at room temperature in the dark for 10 min. Apoptotic cell populations were analyzed using flow cytometry (10,000 events per sample), detecting FITC (FL1) and PI (FL3). Data were analyzed with FlowJo_v10.8.1 software. Live cells were double-negative, early apoptotic cells were FITC-positive/PI-negative, and late apoptotic cells were double-positive [36].

### 2.10. Detection of Mitochondrial Membrane Potential

Mitochondrial membrane potential was assessed using JC-1 dye (Beyotime Biotechnology). Cells were incubated with JC-1 in culture medium at 37 °C for 20 min. After washing twice with PBS, cells were imaged under a fluorescence microscope (Leica DMi8, Wetzlar, Germany) [37].

### 2.11. Mitochondrial Permeability Transition Pore Assay

The mitochondrial permeability transition pore (mPTP) was assessed using the Image-iT™ LIVE Mitochondrial Transition Pore Assay Kit (Invitrogen, I35103, Waltham, MA, USA). A working solution was prepared by mixing calcein-AM, MitoTracker Red CMXRos, Hoechst 33342, and CoCl_2_ in modified HBSS. After staining, cells were imaged under a fluorescence microscope (Leica DMi8) [38].

### 2.12. Mitochondrial ROS (mtROS) Measurement

Mitochondrial superoxide levels were evaluated using MitoSOX™ Red (Invitrogen, M36005, Waltham, MA, USA) according to the manufacturer’s instructions. Cells were incubated with 5 µM MitoSOX for 10 min at 37 °C in the dark, then washed and imaged via fluorescence microscopy [5].

### 2.13. Immunofluorescence

Cells fixed in 4% paraformaldehyde were incubated overnight at 4 °C with primary antibodies, followed by fluorescent secondary antibodies: CoraLite488-conjugated anti-rabbit IgG and CoraLite594-conjugated anti-mouse IgG (Proteintech, Rosemont, Il, USA). DAPI-containing mounting medium (Beyotime, P0131) was used to stain nuclei. Fluorescence signals were captured using a Leica DMi8 microscope [32].

### 2.14. Animal Models

NCG mice (6 weeks old) were used to establish CRC xenograft models. Mice were maintained under SPF conditions with a 12 h light/dark cycle. Each mouse received 2 × 10^6^ CRC cells suspended in 100 µL PBS via subcutaneous injection. Tumor size was measured every 3 days, and volume was calculated using the formula: V (mm^3^) = (L × W^2^)/2, where L is the longest diameter, and W is the shortest [39].

### 2.15. Statistical Analysis

All in vitro experiments were performed at least three times in triplicate. Data are expressed as mean ± SD. Comparisons between the two groups were performed using a two-tailed Student’s *t*-test. Chi-square tests were used for categorical variables. Correlation analysis was conducted using Spearman’s rank method. Kaplan–Meier survival curves were analyzed with the log-rank test. GraphPad Prism 8 was used for all statistical analyses. A *p*-value < 0.05 was considered statistically significant (* *p* < 0.05; ** *p* < 0.01, *** *p* < 0.001).

## 3. Result

### 3.1. DIRAS1 Expression Is Significantly Associated with OXA Resistance and Poor Prognosis in CRC Patients

To identify molecular determinants of OXA resistance in CRC, we collected tumor and adjacent normal tissues from a cohort of 50 CRC patients who subsequently received OXA-based adjuvant chemotherapy. Disease-free survival (DFS) was monitored through long-term follow-up. Patients with recurrence or metastasis were classified as OXA-insensitive (non-responders), while those with sustained DFS were considered OXA-sensitive (responders). From this cohort, five responder and five non-responder cases were selected for transcriptomic profiling. Tissue samples were stratified into four groups: responder tumor (RT), responder normal (RN), non-responder tumor (NRT), and non-responder normal (NRN).

Total RNA was extracted, and transcriptomic analysis was performed. Differentially expressed genes (DEGs) were identified by bioinformatics analysis. Two DEG subsets were generated: genes upregulated in tumor tissues versus matched normal tissues (orange, Figure 1A) and genes upregulated in NRT versus RT samples (blue, Figure 1A). The intersection of these sets yielded candidate genes potentially involved in OXA chemoresistance. Expression profiles of all DEGs are listed in Appendix A. Notably, nine genes exhibited a log2 fold change (log_2_FC) > 4, including DIRAS1, which showed the highest differential expression (log_2_FC = 7.48; Figure 1B).

To functionally validate these candidates, knockdown constructs targeting each of the nine genes were introduced individually into GFP-labeled HCT116 CRC cells. Cell viability assays following OXA exposure demonstrated that DIRAS1 knockdown markedly sensitized cells to OXA (Appendix A), highlighting its potential role in chemoresistance.

WB further confirmed significantly elevated DIRAS1 protein expression in CRC tumor tissues, particularly in NRT samples (Figure 1C and Appendix A). IHC analysis of tumor sections from the 50-patient cohort corroborated these findings, revealing notably higher DIRAS1 expression in non-responders compared to responders (Figure 1D and Appendix A). Correlational analysis between DIRAS1 levels and patient outcomes based on telephone follow-up (Appendix A) demonstrated that high DIRAS1 expression was associated with significantly reduced DFS and overall survival (Figure 1E).

These results strongly implicate DIRAS1 as a key factor in CRC chemoresistance. DIRAS1 may serve as a predictive biomarker and potential therapeutic target for overcoming platinum-based treatment failure in CRC.

### 3.2. DIRAS1 Enhances Proliferation and Migration in CRC Cells

DIRAS1 has been implicated in the regulation of tumor cell proliferation, migration, and chemoresistance in multiple solid malignancies [40]. To elucidate its function in CRC, we first assessed endogenous DIRAS1 expression at both the mRNA and protein levels across three CRC cell lines—HCT116, DLD1, and SW620—using RT-PCR and WB. Results revealed significantly higher DIRAS1 expression in HCT116 and SW620 cells compared to DLD1 cells (Figure 2A,B and Appendix A). Based on these findings, DLD1 cells were selected for generating stable DIRAS1-overexpressing (OE-DIRAS1) lines, while HCT116 cells were used to establish both DIRAS1-knockdown (Sh-DIRAS1) and OE-DIRAS1 models. The efficiency of gene modulation was confirmed by WB analysis (Figure 2C,D and Appendix A).

Subsequent functional assays, including MTT, colony formation, and wound healing assays, were conducted to evaluate the phenotypic consequences of DIRAS1 modulation. In HCT116 cells, DIRAS1 knockdown significantly inhibited cell proliferation, as evidenced by reduced cell viability, smaller colony size, and decreased colony number (Figure 2E,H). Furthermore, wound healing assays demonstrated that DIRAS1 silencing markedly impaired the migratory capacity of these cells (Figure 2K).

Conversely, DIRAS1 overexpression enhanced both proliferative and migratory behavior. OE-DIRAS1 cells exhibited significantly increased cell viability and colony formation in both HCT116 and DLD1 models (Figure 2F,G,I,J). Consistent with these findings, wound healing assays showed that DIRAS1 overexpression significantly accelerated cell migration in both cell lines (Figure 2L,M).

Collectively, these data suggest that DIRAS1 functions as a pro-oncogenic factor in CRC by promoting tumor cell proliferation and migration.

### 3.3. Overexpression of DIRAS1 Promotes OXA Resistance in CRC Cells In Vitro

To elucidate the role of DIRAS1 in OXA resistance, HCT116 CRC cells were treated with 2 µM OXA and harvested at multiple time points (0, 12, 24, 48, 96, 120, 144, and 168 h) to monitor DIRAS1 expression dynamics. RT-PCR and WB analyses revealed a time-dependent upregulation of DIRAS1 at both mRNA and protein levels. Notably, DIRAS1 protein levels increased nearly threefold after 48 h of OXA treatment compared to untreated controls (*p* < 0.05) (Figure 3A,B). A similar response was observed in SW620 cells, whereas no effect was detected in inherently DIRAS1-negative DLD1 cells (Appendix A), suggesting cell-type specificity.

To validate the functional significance of DIRAS1, stable DIRAS1 knockdown (Sh-DIRAS1) and overexpressing (OE-DIRAS1) HCT116 cell lines were established. Fluorescence microscopy-based cell viability assays showed that Sh-DIRAS1 cells displayed a significantly lower survival rate than control cells following 48 h of OXA exposure (8% vs. 33%; *p* < 0.01) (Figure 3C,D).

Further evaluation of chemosensitivity was conducted by exposing cells to escalating concentrations of OXA (0–60 µM) for 24 h to calculate IC50 values. OE-DIRAS1 cells exhibited a significantly higher IC50 (17.12 µM) than control cells (5.94 µM; *p* < 0.01), indicating enhanced drug resistance. In contrast, Sh-DIRAS1 cells showed a markedly reduced IC50 (2.92 µM; *p* < 0.05) (Figure 3E,F), suggesting an increased sensitivity to OXA.

To investigate the potential involvement of apoptosis, Annexin V/propidium iodide (PI) dual-staining flow cytometry was performed. Treatment with 10 µM OXA for 48 h significantly elevated the apoptosis rate in Sh-DIRAS1 cells compared to controls (*p* < 0.05), while OE-DIRAS1 cells exhibited a slight but reproducible reduction in apoptosis under the same conditions (Figure 3G,H).

These results demonstrate that DIRAS1 confers a robust chemoprotective effect in CRC cells. Overexpression of DIRAS1 markedly enhances resistance to OXA, whereas its downregulation restores drug sensitivity, potentially via modulation of the apoptotic pathway.

### 3.4. DIRAS1 Confers OXA Resistance in a CRC Xenograft Model

To further validate the role of DIRAS1 in OXA resistance under in vivo conditions, we established a CRC xenograft model using NCG mice. HCT116 cells stably expressing either DIRAS1 shRNA (Sh-DIRAS1) or control shRNA were subcutaneously implanted. Upon tumor volumes reaching ~100 mm^3^, mice were administered intravenous OXA (5 mg/kg) on days 1, 5, and 9. Tumor growth was monitored every three days for five weeks.

Tumors derived from Sh-DIRAS1 cells displayed significantly slower growth rates and smaller final volumes compared to controls (Figure 4A–C, *p* < 0.05). Notably, the Sh-DIRAS1 group showed markedly enhanced sensitivity to OXA treatment, as evidenced by a greater reduction in tumor burden. Gross imaging and tumor weight measurements corroborated the reduction in tumor mass. Furthermore, IHC analysis of xenograft tissues confirmed effective suppression of DIRAS1 expression in Sh-DIRAS1 tumors relative to controls (Figure 4D and Appendix A).

These findings suggest that DIRAS1 functions as a key mediator of platinum resistance in CRC and may represent a viable therapeutic target to improve clinical responses to OXA-based chemotherapy.

### 3.5. PHB1 Is a Downstream Target of DIRAS1 That Mediates OXA Resistance

To delineate the downstream effectors through which DIRAS1 mediates OXA resistance in CRC, we performed high-throughput RNA sequencing in DIRAS1-overexpressing (OE-DIRAS1) and control HCT116 cells. Differential gene expression analysis identified Prohibitin 1 (PHB1) as significantly upregulated in OE-DIRAS1 cells (Figure 5A, Appendix A), suggesting a potential regulatory relationship. Validation experiments confirmed that PHB1 expression was positively regulated by DIRAS1: DIRAS1 overexpression led to increased PHB1 protein levels, whereas DIRAS1 knockdown suppressed PHB1 expression (Figure 5B and Appendix A). Conversely, altering PHB1 levels via overexpression or knockdown had no significant effect on DIRAS1 expression (Figure 5C), indicating a unidirectional regulatory axis from DIRAS1 to PHB1.

To evaluate the functional contribution of PHB1 to OXA resistance, we assessed cell viability under graded OXA concentrations. PHB1 overexpression significantly elevated the IC_50_ from 5.64 µM to 15.32 µM (*p* < 0.01), while PHB1 knockdown markedly reduced the IC_50_ from 6.12 µM to 2.86 µM (*p* < 0.05) in HCT116 cells (Figure 5D,E), confirming that PHB1 promotes chemoresistance.

To further examine the DIRAS1–PHB1 axis in regulating apoptosis under chemotherapeutic stress, rescue experiments were conducted. DIRAS1 knockdown significantly increased late apoptosis upon OXA exposure, which was effectively reversed by PHB1 overexpression (Sh-DIRAS1 + OE-PHB1 group) (Figure 5F). Similarly, PHB1 knockdown sensitized cells to apoptosis, and this effect was only partially reversed by DIRAS1 overexpression (Sh-PHB1 + OE-DIRAS1 group) (Figure 5G). Moreover, simultaneous knockdown of both DIRAS1 and PHB1 synergistically increased apoptosis, while co-overexpression of both genes suppressed it (Appendix A), indicating a functional interdependence between DIRAS1 and PHB1 in apoptotic regulation.

To validate these findings in clinical samples, Western blot analysis was performed on paired CRC tumors and adjacent normal tissues. PHB1 protein levels were significantly upregulated in tumor tissues compared to matched normal counterparts (*p* < 0.05; *n* = 4 pairs) (Figure 6A,C). Moreover, DIRAS1 and PHB1 protein levels were positively correlated in tumor samples, with linear regression yielding an R^2^ of 0.7857 (*p* = 0.0480) (Figure 6B,D). Immunofluorescence staining further confirmed the co-enrichment of DIRAS1 and PHB1 in CRC tumor tissues relative to adjacent and distant normal tissues (Figure 6E).

Together, these results identify PHB1 as a direct downstream effector of DIRAS1 and a key contributor to OXA resistance in CRC. While PHB1 overexpression is capable of mitigating apoptosis induced by DIRAS1 knockdown, the partial reversal of apoptosis in PHB1-deficient cells overexpressing DIRAS1 indicates that PHB1 is necessary but not solely sufficient for DIRAS1-mediated chemoprotection. These findings highlight the DIRAS1–PHB1 axis as a critical pathway in CRC chemoresistance and warrant further mechanistic exploration of its downstream signaling components.

### 3.6. DIRAS1 Regulates PHB1-Mediated Mitochondrial Homeostasis

PHB1 is a multifunctional mitochondrial scaffold protein critical for maintaining mitochondrial integrity, participating in mitochondrial maturation, protein quality control, and inner membrane stabilization [41,42]. To determine whether the DIRAS1–PHB1 axis influences mitochondrial function, we performed Kyoto Encyclopedia of Genes and Genomes (KEGG) pathway enrichment analysis on genes upregulated in DIRAS1-overexpressing (OE-DIRAS1) HCT116 cells. The OXPHOS pathway emerged as the top enriched pathway (Figure 7A), suggesting a potential regulatory role for DIRAS1 in mitochondrial metabolism.

Based on these findings, we assessed mitochondrial membrane potential (ΔΨm) using JC-1 staining in three HCT116 cell groups: control (non-targeting shRNA + empty vector), DIRAS1 knockdown (Sh-DIRAS1 + empty vector), and DIRAS1 knockdown with PHB1 overexpression (Sh-DIRAS1 + OE-PHB1). DIRAS1 knockdown significantly reduced ΔΨm, indicated by a decrease in the red/green fluorescence ratio. This reduction was restored upon PHB1 overexpression (Figure 7B,D). Flow cytometry further validated these results (Figure 7C,E).

Next, we evaluated the mitochondrial permeability transition pore (mPTP) opening using the calcein-AM/cobalt quenching assay. Sh-DIRAS1 cells displayed increased mPTP opening, reflecting compromised mitochondrial membrane integrity. PHB1 co-overexpression reversed this phenotype (Figure 7F,I). Additionally, mitochondrial reactive oxygen species (mtROS) levels, detected via MitoSOX Red staining, were significantly elevated in DIRAS1-silenced cells, and this increase was attenuated by PHB1 restoration (Figure 7G,H).

Together, these findings indicate that DIRAS1 maintains mitochondrial homeostasis via transcriptional upregulation of PHB1. Specifically, DIRAS1 enhances mitochondrial membrane potential, restricts mPTP opening, and reduces mtROS accumulation, each of which contributes to protecting CRC cells against OXA-induced mitochondrial damage.

## 4. Discussion

Our study demonstrates a key mechanism of DIRAS1 in chemotherapy resistance of CRC and proposes a novel molecular pathway that promotes tumor cell resistance by regulating PHB1 to maintain mitochondrial homeostasis, which provides new theoretical basis and potential clinical strategies for targeted drug resistance. DIRAS1, a small GTPase within the Ras superfamily, has been previously reported to function as a tumor suppressor in various human malignancies [14]. In esophageal squamous cell carcinoma (ESCC), reduced DIRAS1 expression correlates with advanced clinical stage, lymph node metastasis, and poor overall survival. Mechanistically, DIRAS1 has been shown to enhance apoptosis and suppress metastasis via modulation of the ERK1/2 and p38 MAPK signaling pathways [43]. Similar tumor-suppressive functions have been observed in cervical cancer, where DIRAS1 downregulation facilitates tumor progression while its overexpression inhibits proliferation, migration, and invasion [18]. Moreover, treatment with DNA methylation and histone deacetylase inhibitors increases DIRAS1 mRNA levels without affecting protein abundance, suggesting complex, transcriptionally centered epigenetic regulation.

In contrast, our findings highlight a distinct oncogenic role for DIRAS1 in CRC. Our transcriptomic and functional analyses reveal that DIRAS1 is significantly upregulated in OXA-insensitive CRC tissues. In vitro and in vivo assays confirm that DIRAS1 promotes CRC cell proliferation, migration, and resistance to OXA. Furthermore, survival analysis demonstrates that high DIRAS1 expression is strongly associated with adverse clinical outcomes.

This functional divergence suggests that DIRAS1′s role in CRC is context-dependent, governed by multifaceted regulatory mechanisms. While promoter methylation-mediated suppression represents a well-established paradigm [44], our data suggest that in specific CRC subtypes, particularly those with acquired chemoresistance, DIRAS1 is upregulated and exerts pro-tumorigenic functions. Such a shift could be driven by aberrant activation of upstream signaling cascades, post-transcriptional modulation, changes in protein turnover, or context-specific epigenetic reprogramming beyond promoter methylation [40]. The observed discrepancy between mRNA and protein expression in cervical cancer further supports a multilayered regulation model [18]. Therefore, the oncogenic role of DIRAS1 in CRC, as demonstrated in our model, likely results from context-specific regulatory overrides, emphasizing the need for deeper exploration into the signaling pathways and post-translational mechanisms dictating DIRAS1′s dual functionality.

OXA, a third-generation platinum-based chemotherapeutic, remains a cornerstone of adjuvant therapy in stage III CRC [7]. Its combination with fluoropyrimidines has been shown to significantly improve overall survival and reduce recurrence rates in patients following curative resection [45]. Mechanistically, OXA induces G2/M cell cycle arrest and apoptosis in CRC cells [10], involving Bax translocation to mitochondria, cytochrome c release, and Caspase-3 activation [9,46]. Importantly, our findings establish that DIRAS1 transcriptionally upregulates Prohibitin 1 (PHB1). The DIRAS1–PHB1 axis maintains mitochondrial membrane potential and integrity, thereby reducing OXA-induced apoptotic responses.

PHB1 is known to preserve mitochondrial cristae morphology and membrane potential [4,47,48], as well as support assembly of respiratory chain complexes [4]. Its overexpression has been implicated in chemoresistance in various malignancies [49], but its role in OXA resistance within CRC has not been previously elucidated. Our demonstration of unidirectional transcriptional regulation of PHB1 by DIRAS1 provides a novel mechanistic link between an oncogene and mitochondrial stabilization as a chemoprotective strategy. Considering that the cytotoxic effects of OXA are partially dependent on the induction of apoptosis via mitochondrial pathways [5], this DIRAS1-driven stabilization of mitochondrial membranes may directly suppress apoptotic signaling. Furthermore, our observation of DIRAS1 upregulation in response to prolonged OXA exposure highlights its adaptive function in the development of acquired resistance. Collectively, the DIRAS1–PHB1–mitochondrial integrity axis constitutes a novel and mechanistically distinct pathway, differing from previously characterized canonical resistance mechanisms.

Despite these insights, several limitations should be acknowledged. First, our reliance on subcutaneous xenograft models may not fully recapitulate the tumor microenvironment, which is better preserved in orthotopic or patient-derived xenograft (PDX) models [50]. Second, while we have established that DIRAS1 upregulates PHB1 at the transcriptional level, the precise molecular mechanism, whether through direct promoter binding or via intermediary transcription factors, remains unclear and warrants further investigation. Third, although our data demonstrate that PHB1 contributes to mitochondrial membrane stability and resistance, the downstream mitochondrial targets of the DIRAS1–PHB1 axis, such as specific alterations in oxidative phosphorylation (OXPHOS), electron transport chain function, ROS balance, or cristae architecture, remain to be fully characterized [51]. Finally, validation of our findings in larger cohorts of CRC patient samples, especially in pre- and post-OXA therapy contexts, is essential to establish the clinical utility of DIRAS1 and PHB1 as predictive biomarkers.

Based on our findings, several future research directions are proposed. First, DIRAS1 may regulate additional mitochondrial effectors beyond PHB1, collectively contributing to chemoresistance. Mitochondrial proteomic profiling of DIRAS1-modified cells may help identify these targets. Second, considering that PHB1 is known to prevent mitochondrial permeability transition pore (mPTP) opening, a key step in apoptosis, direct examination of mPTP kinetics could clarify the DIRAS1–PHB1 protective mechanism. Third, given PHB1′s role in supporting complex I assembly and OXPHOS, alterations in metabolic flux and ROS levels could be assessed via metabolomic and redox assays. Moreover, combinatorial in vivo therapy targeting the DIRAS1–PHB1 axis alongside OXA, perhaps through PHB1 inhibitors or RNAi strategies, could be investigated [52,53]. Finally, elucidating upstream signals driving DIRAS1 overexpression under chronic OXA pressure may reveal early events in resistance development and provide additional therapeutic opportunities.

## 5. Conclusions

This study provides critical insights into the role of DIRAS1 in CRC chemoresistance and elucidates its underlying molecular mechanisms. Our findings demonstrate that DIRAS1 is significantly overexpressed in CRC tissues and cell lines, with its expression closely correlated with OXA resistance and poor patient prognosis. Both in vitro and in vivo experiments confirmed that DIRAS1 enhances CRC cell proliferation, migration, and chemoresistance. Mechanistically, we identified PHB1 as a transcriptional target of DIRAS1, revealing that DIRAS1 contributes to chemoresistance by preserving mitochondrial function through upregulation of PHB1. This regulatory axis stabilizes mitochondrial membrane potential and integrity, protecting CRC cells from OXA-induced apoptosis. Targeting this pathway could restore mitochondrial susceptibility to OXA-induced damage, ultimately enhancing the efficacy of this cornerstone chemotherapeutic and improving clinical outcomes in advanced CRC.

## Figures and Tables

**Figure 1 biology-14-00819-f001:**
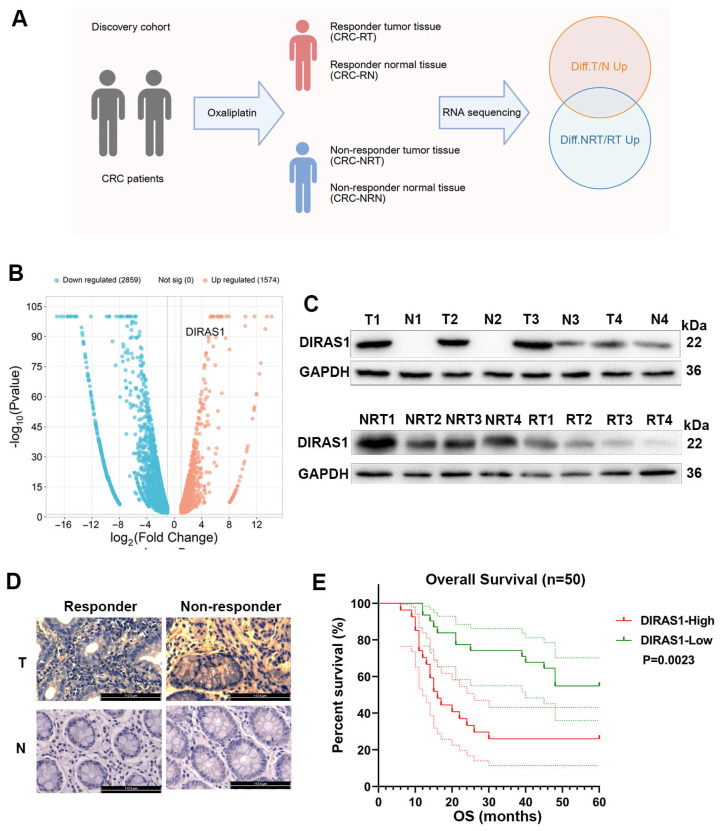
Elevated DIRAS1 expression in clinical CRC samples is associated with oxaliplatin resistance and poor prognosis. (**A**) Schematic overview of transcriptomic analysis and differential gene expression (DEG) identification. Tumor and matched adjacent normal tissues were collected from 10 CRC patients receiving OXA-based adjuvant chemotherapy. Based on postoperative treatment response, patients were classified as responders (no recurrence/metastasis) or non-responders (recurrence/metastasis), and samples were divided into four groups: responder tumor (RT), responder normal (RN), non-responder tumor (NRT), and non-responder normal (NRN) (*n* = 5 per group). mRNA was extracted and subjected to high-throughput RNA sequencing. DEGs were identified with thresholds of |log_2_FC| > 1 and adjusted *p* < 0.05. Orange circles denote genes upregulated in tumor tissue (RT + NRT) vs. normal tissue (RN + NRN), and blue circles represent genes upregulated in NRT vs. RT. The intersection of these two sets revealed candidate genes associated with OXA resistance. (**B**) Volcano plot showing DEGs between NRT and RT samples. Genes significantly upregulated in NRT are shown in orange, and those downregulated are in blue (|log_2_FC| > 1, adjusted *p* < 0.05). Among the nine most differentially expressed genes (log_2_FC > 4), DIRAS1 showed the highest upregulation (log_2_FC = 7.48, indicated). (**C**) WB analysis of DIRAS1 expression in paired tumor (T) and normal (N) tissues from responder (R) and non-responder (NR) patients (*n* = 4 per group). GAPDH was used as a loading control. Densitometric quantification (mean ± SEM) indicated significantly higher DIRAS1 protein levels in tumor tissues, particularly in NRT. (unpaired Student’s *t*-test). (**D**) IHC staining of DIRAS1 in CRC tissue sections from responder and non-responder patients (*n* = 50). Representative images and H-score quantification show markedly increased DIRAS1 expression in NRT compared to RT tissues. Scale bar: 112.5 µm. Statistical significance was determined using the Mann–Whitney U test. (**E**) Kaplan–Meier analysis of overall survival (OS) in 50 CRC patients stratified by DIRAS1 expression levels (high vs. low, based on median IHC H-score). Patients with high DIRAS1 expression had significantly worse OS (log-rank test). Median OS: high DIRAS1 = 15 months; low DIRAS1 = 48 months.

**Figure 2 biology-14-00819-f002:**
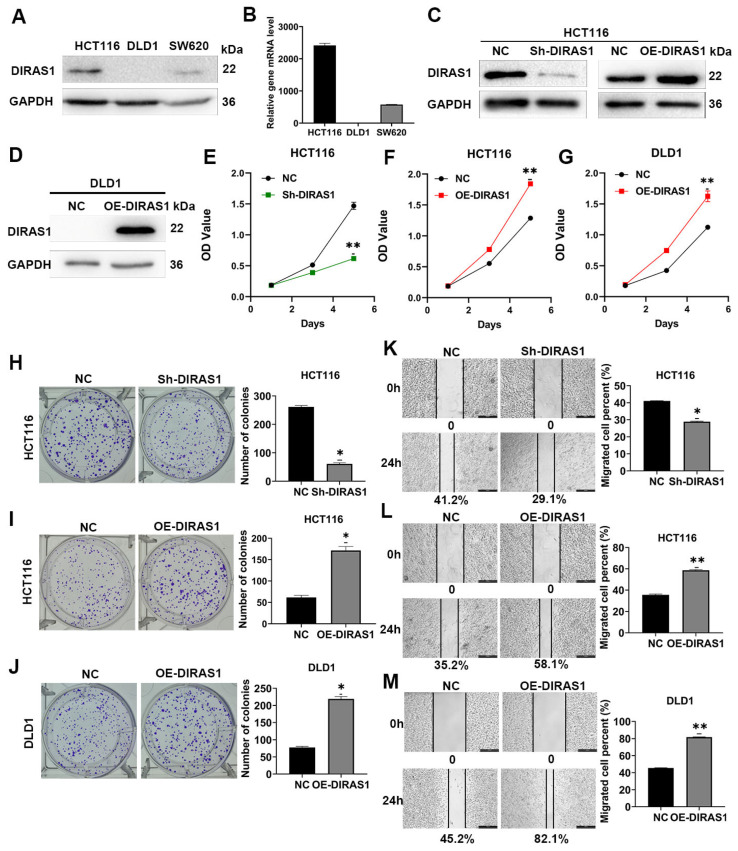
DIRAS1 enhances proliferation and migration in CRC cells. (**A**) WB analysis of endogenous DIRAS1 protein expression in HCT116, DLD1, and SW620 CRC cell lines. (**B**) Quantitative analysis of DIRAS1 mRNA expression in HCT116, DLD1, and SW620 cells. Data represent mean ± SD from three independent experiments. (**C**) Validation of DIRAS1 knockdown and overexpression in HCT116 cells by WB analysis. (**D**) WB confirmation of DIRAS1 overexpression in DLD1 cells. (**E**–**G**) MTT assays measuring cell proliferation: (**E**) DIRAS1 knockdown significantly reduced proliferation in HCT116 cells. (**F**) DIRAS1 overexpression significantly enhanced proliferation in HCT116 cells. (**G**) DIRAS1 overexpression significantly enhanced proliferation in DLD1 cells. All data are shown as mean ± SD (*n* = 3 independent experiments, each in triplicate). Statistical significance was assessed using an unpaired two-tailed Student’s *t*-test (** *p* < 0.01). (**H**–**J**) Colony formation assays: (**H**) DIRAS1 knockdown reduced colony formation ability in HCT116 cells (12-day culture). (**I**) DIRAS1 overexpression increased colony number and size in HCT116 cells (10-day culture). (**J**) DIRAS1 overexpression enhanced colony formation in DLD1 cells (10-day culture). Representative images and quantification are shown. Data represent mean ± SD (*n* = 3 independent experiments). Statistical significance: * *p* < 0.05. (**K**–**M**) Wound healing assays to assess cell migration: (**K**) DIRAS1 knockdown significantly impaired migratory capacity of HCT116 cells at 24 h post-scratch. (**L**) DIRAS1 overexpression significantly enhanced migration in HCT116 cells. (**M**) DIRAS1 overexpression significantly enhanced migration in DLD1 cells. Representative phase-contrast images at 0 h and 24 h are shown (scale bar = 450 µm), with quantified wound closure rates (mean ± SD, *n* = 3 independent experiments). Statistical significance: * *p* < 0.05, ** *p* < 0.01 by unpaired *t*-test.

**Figure 3 biology-14-00819-f003:**
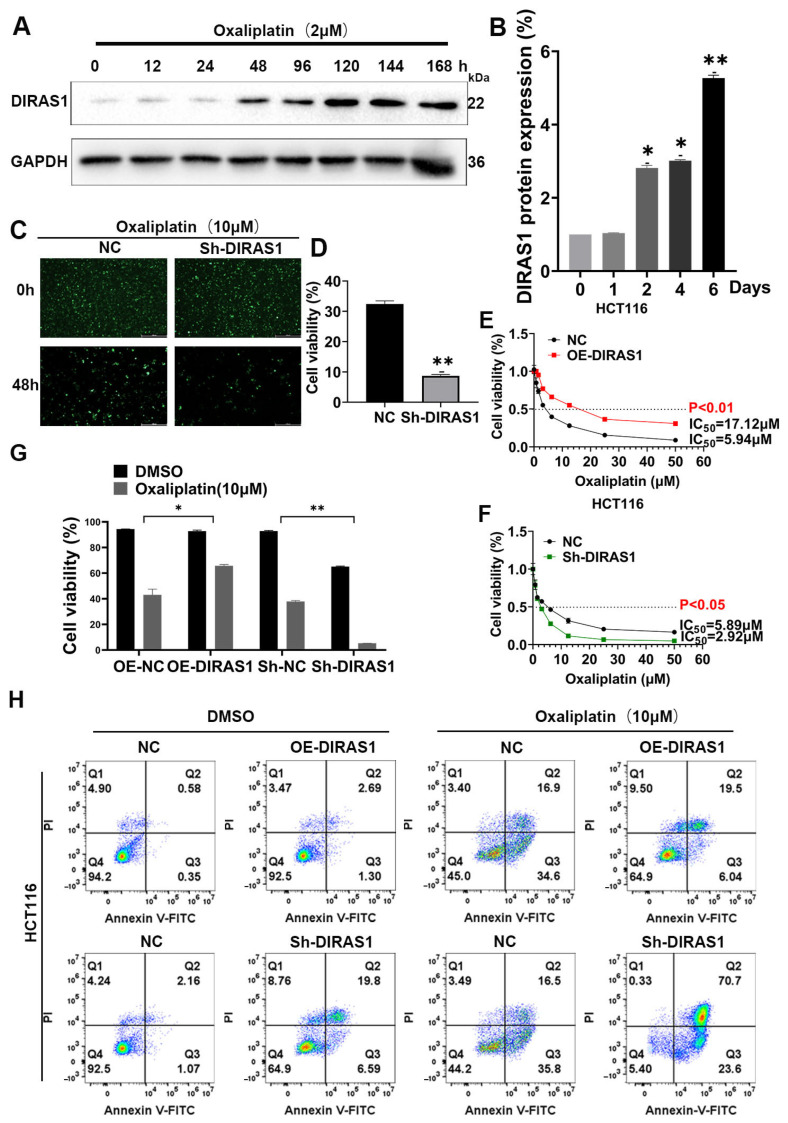
Overexpression of DIRAS1 promotes oxaliplatin resistance in CRC cells in vitro. (**A**,**B**) DIRAS1 expression is induced by OXA in a time-dependent manner. (**A**) WB analysis of DIRAS1 expression in HCT116 cells treated with 2 µM OXA for indicated durations (0–168 h). GAPDH served as a loading control. (**B**) Quantification of DIRAS1 band intensity normalized to GAPDH (mean ± SD, *n* = 3 independent experiments) (* *p* < 0.05, ** *p* < 0.01). (**C**,**D**) Cell viability assessment by GFP fluorescence following OXA exposure. (**C**) Representative fluorescence microscopy images of GFP-labeled HCT116 cells (control, Sh-DIRAS1, OE-DIRAS1) treated with 2 µM OXA for 48 h. Scale bar: 450 µm. Viable cells display green fluorescence. (**D**) Quantification of viable GFP-positive cells from (C). Data represent mean ± SD (*n* = 3 independent experiments). Unpaired two-tailed Student’s *t*-test (** *p* < 0.01). (**E**,**F**) DIRAS1 modulates OXA sensitivity in CRC cells. (**E**) MTT assays showing dose–response curves of HCT116 cells (control, Sh-DIRAS1, OE-DIRAS1) treated with increasing concentrations of OXA (0–60 µM) for 24 h. (**F**) IC_50_ values calculated from dose–response data (mean ± SD, *n* = 3). Knockdown of DIRAS1 significantly reduced, while overexpression increased the IC_50_ for OXA (* *p* < 0.05, ** *p* < 0.01; unpaired *t*-test). (**G**,**H**) DIRAS1 influences apoptosis under OXA treatment. (**G**) Quantification of viable (Annexin V^−^/PI^−^) cells from (**H**). Data are presented as mean ± SD (*n* = 3 independent experiments). Statistical significance determined by unpaired *t*-test (* *p* < 0.05, ** *p* < 0.01). (H) Representative flow cytometry plots of Annexin V-FITC/PI dual staining in HCT116 cells (control, Sh-DIRAS1, OE-DIRAS1) treated with 10 µM OXA for 48 h.

**Figure 4 biology-14-00819-f004:**
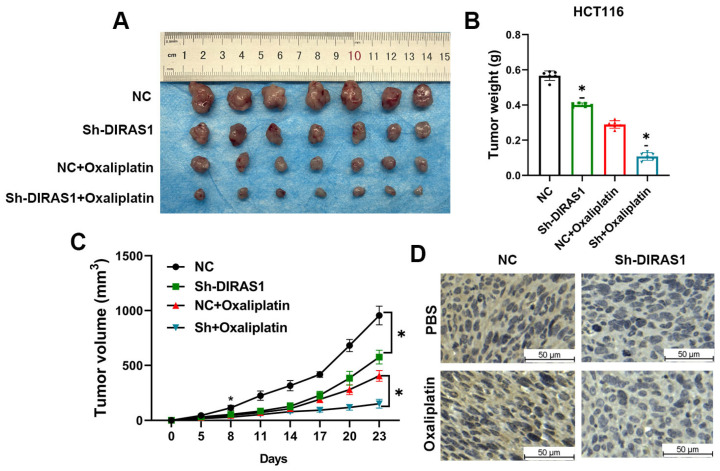
DIRAS1 confers oxaliplatin resistance in a CRC xenograft model. (**A**) Nude mice were subcutaneously injected with HCT116 cells stably expressing either control shRNA or DIRAS1-targeting shRNA (shDIRAS1). Upon tumor establishment (~100 mm^3^), mice were randomly assigned to receive intravenous OXA (5 mg/kg) or PBS on days 1, 5, and 9 (*n* = 7 mice/group). Tumor volumes were measured every 3 days. Experimental groups: control (vehicle), control + OXA, ShDIRAS1 (vehicle), ShDIRAS1 + OXA. (**B**) Final tumor weight at the endpoint. Tumors were harvested and weighed. Data are presented as mean ± SEM (*n* = 7). OXA treatment significantly reduced tumor weight in control mice (0.58 ± 0.12 g to 0.36 ± 0.08 g, * *p* < 0.05), and a more pronounced reduction was observed in the ShDIRAS1 group (0.40 ± 0.03 g to 0.10 ± 0.02 g, * *p* < 0.05 vs. control + OXA). Statistical analysis: two-way ANOVA with Tukey’s post hoc test. (**C**) Tumor growth kinetics. Tumor volume progression over time was plotted for each group. Data are shown as mean ± SEM (*n* = 7). Tumor growth was significantly delayed in the ShDIRAS1 + OXA group compared to control + OXA at the experimental endpoint (* *p* < 0.05; repeated measures two-way ANOVA with Šidák’s post hoc test). (**D**) Validation of DIRAS1 knockdown by IHC. Representative IHC staining images of DIRAS1 expression in excised tumor tissues from each group. Scale bar: 50 µm. Semi-quantitative H-scoring revealed significantly decreased DIRAS1 expression in ShDIRAS1 tumors compared to controls (* *p* < 0.05; Mann–Whitney U test, *n* = 7 tumors/group). Data are shown as mean H-score ± SEM.

**Figure 5 biology-14-00819-f005:**
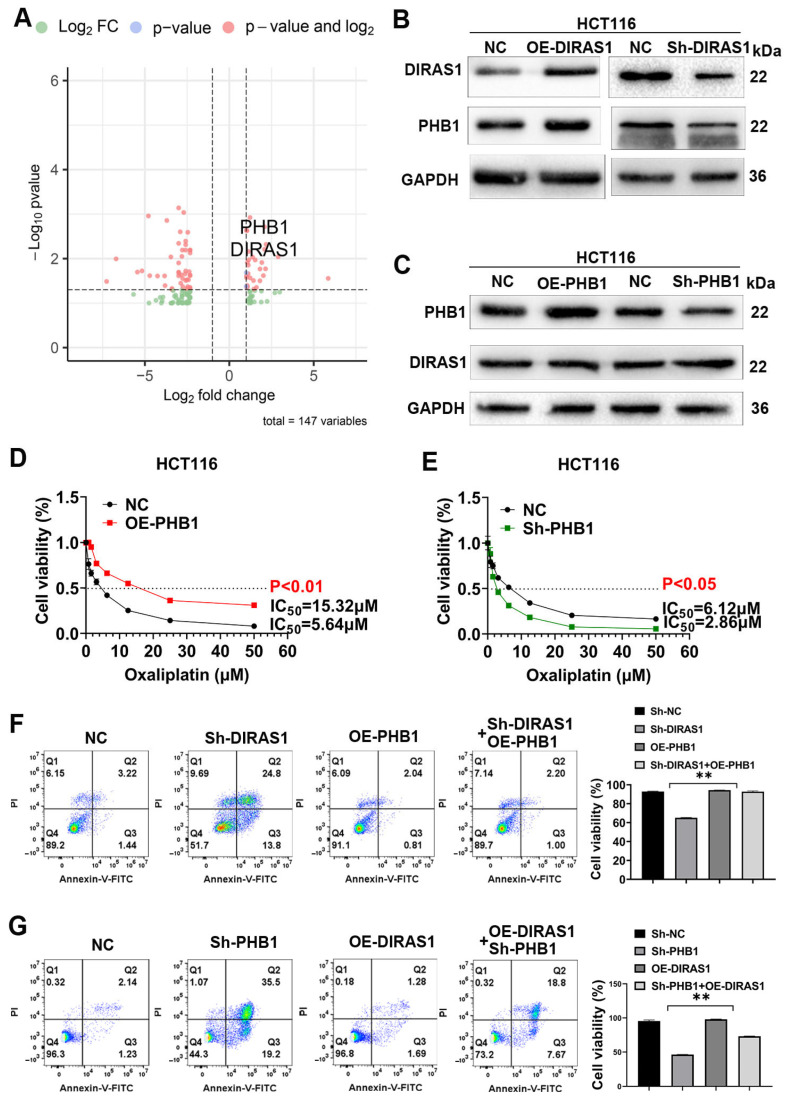
PHB1 is a downstream effector of DIRAS1 that mediates oxaliplatin resistance in CRC cells. (**A**) Transcriptomic identification of DIRAS1-regulated genes. Volcano plot showing DEGs in HCT116 cells overexpressing DIRAS1 (OE-DIRAS1) compared to control, based on high-throughput RNA-seq. Screening criteria: |log_2_FC| > 1, adjusted *p* < 0.05. PHB1 (highlighted) was significantly upregulated (log_2_FC = 2.983, *p* < 0.05). Upregulated and downregulated genes are shown in red and blue, respectively. (**B**) DIRAS1 regulates PHB1 protein expression. WB analysis of DIRAS1 and PHB1 protein levels in HCT116 cells following DIRAS1 knockdown (Sh-DIRAS1) or overexpression (OE-DIRAS1). GAPDH was used as a loading control. Densitometric analysis confirmed a positive correlation between DIRAS1 and PHB1 protein levels (mean ± SD, *n* = 3). (**C**) PHB1 does not regulate DIRAS1 expression. WB analysis showing DIRAS1 levels in HCT116 cells upon PHB1 knockdown (Sh-PHB1) or overexpression (OE-PHB1) compared to control. GAPDH loading control included. Densitometry revealed no significant change in DIRAS1 levels (*p* > 0.05, unpaired *t*-test), indicating unidirectional DIRAS1→PHB1 regulation. (**D**) PHB1 overexpression promotes OXA resistance. MTT assay showing dose-response curves of OE-PHB1 and control HCT116 cells treated with gradient concentrations of OXA (0–60 µM, 24 h). Calculated IC_50_: 15.32 µM for OE-PHB1 vs. 5.64 µM for control (*p* < 0.01, unpaired *t*-test, *n* = 3). (**E**) PHB1 knockdown sensitizes CRC cells to OXA. MTT assay showing enhanced OXA sensitivity in Sh-PHB1 HCT116 cells vs. control. Calculated IC_50_: 2.86 µM (Sh-PHB1) vs. 6.12 µM (control) (*p* < 0.05, *n* = 3, triplicates). Data shown as mean ± SD. (**F**) PHB1 overexpression rescues DIRAS1 knockdown-induced apoptosis. Flow cytometric analysis of apoptosis in HCT116 cells treated with 10 µM OXA for 48 h. Experimental groups: (1) control; (2) Sh-DIRAS1; (3) OE-PHB1; (4) Sh-DIRAS1 + OE-PHB1. Late apoptotic cells quantified as Annexin V^+^/PI^+^ population; viable cells quantified as Annexin V^−^/PI^−^. PHB1 overexpression partially reversed apoptosis induced by DIRAS1 knockdown (** *p* < 0.01, *n* = 3). (**G**) DIRAS1 partially rescues PHB1 knockdown-induced apoptosis. Same experimental design as (**F**) but with PHB1 knockdown and DIRAS1 overexpression. Groups: (1) control; (2) Sh-PHB1; (3) OE-DIRAS1; (4) Sh-PHB1 + OE-DIRAS1. DIRAS1 overexpression partially restored cell viability in PHB1-silenced cells (** *p* < 0.01, *n* = 3).

**Figure 6 biology-14-00819-f006:**
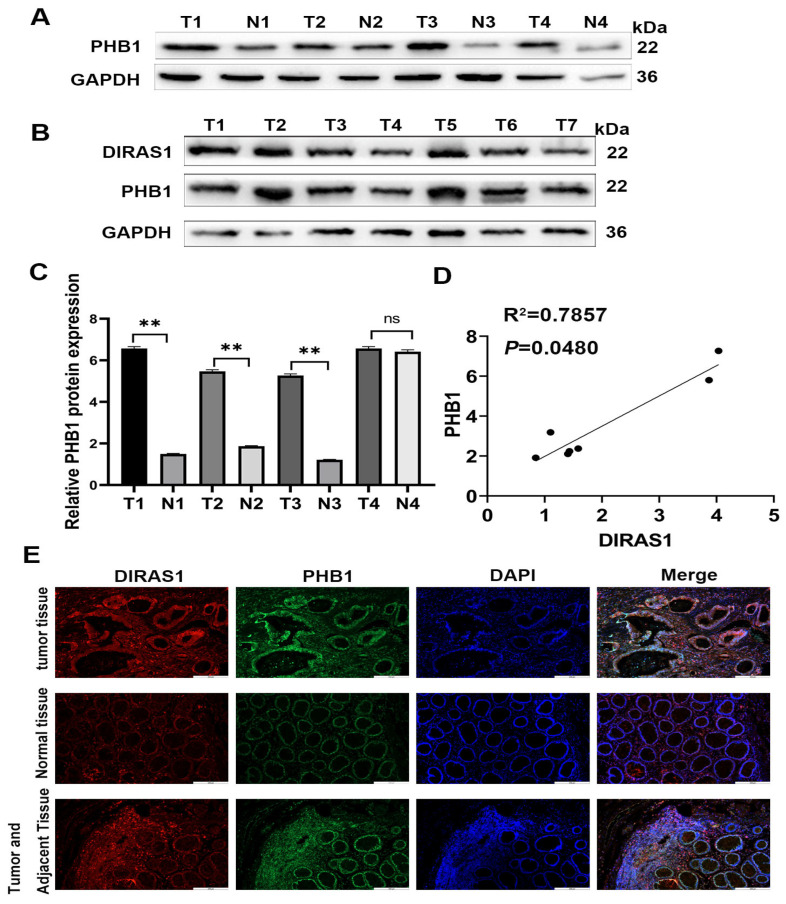
Correlation of DIRAS1 and PHB1 expression in clinical CRC tissues. (**A**) PHB1 protein expression in CRC tumor vs. normal tissues. Representative Western blot images showing PHB1 protein levels in paired CRC tumor (T) and adjacent normal (N) tissues (*n* = 4 pairs). GAPDH served as the loading control. (**B**) Co-expression analysis of DIRAS1 and PHB1 in CRC tumors. WB analysis of DIRAS1 and PHB1 protein levels in individual CRC tumor tissue lysates (*n* = 7). GAPDH loading control included. (**C**) Quantitative analysis of PHB1 expression in paired tissues. Densitometric quantification of PHB1 protein levels from panel (**A**), normalized to GAPDH. Data are presented as mean ± SEM (*n* = 4 pairs). Statistical analysis by paired two-tailed Student’s *t*-test (*** p* < 0.01, ns: *p* > 0.05, tumor vs. normal). (**D**) Positive correlation between DIRAS1 and PHB1 protein levels. Densitometric data from panel (**B**), normalized to GAPDH, were subjected to Pearson correlation analysis. Linear regression revealed a significant positive correlation between DIRAS1 and PHB1 expression (R^2^ = 0.7857, *p* = 0.0480). (**E**) Co-enrichment of DIRAS1 and PHB1 in CRC tissues by immunofluorescence. Representative confocal microscopy images of CRC tumor (T), adjacent normal (AN), and distal normal (N) tissue sections. DIRAS1 was visualized using a CY3-conjugated secondary antibody (red), PHB1 using a FITC-conjugated secondary antibody (green), and nuclei were stained with DAPI (blue). Scale bar = 225 µm.

**Figure 7 biology-14-00819-f007:**
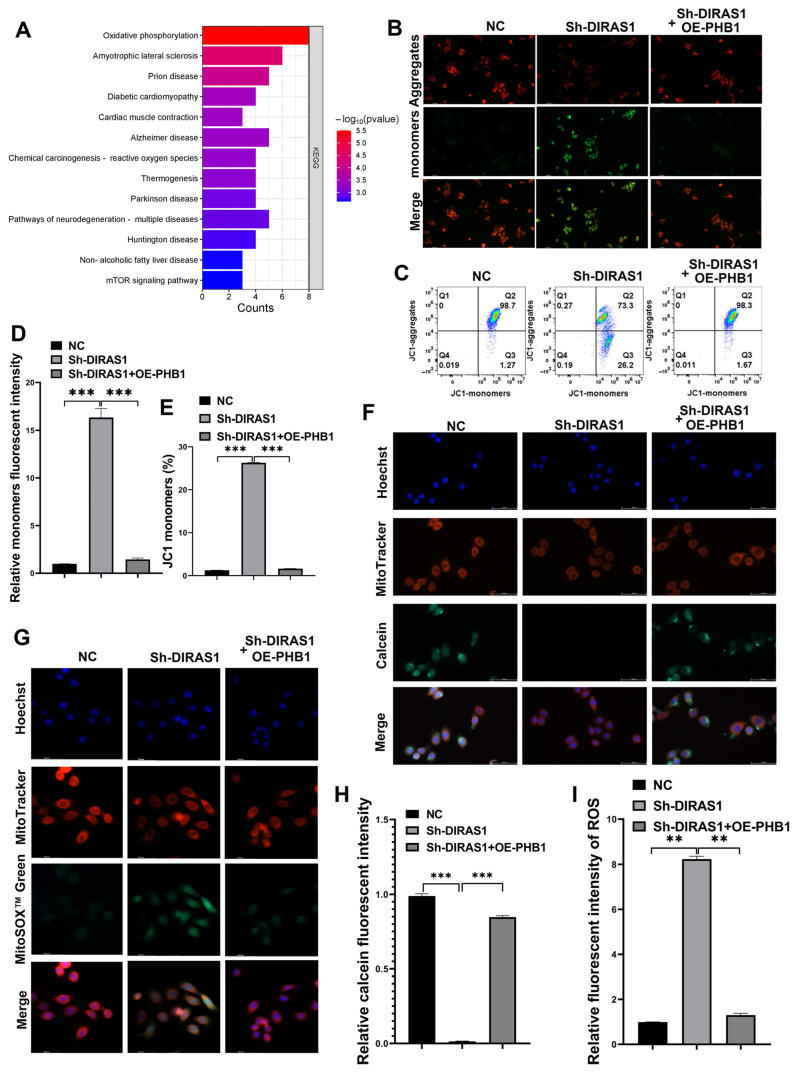
DIRAS1 regulates PHB1-mediated mitochondrial homeostasis in CRC cells. (**A**) KEGG pathway enrichment analysis of DIRAS1-regulated genes. Enrichment analysis of DEGs (|log_2_FC| > 1.5, *p*-adj < 0.05) from RNA sequencing of DIRAS1-overexpressing (OE-DIRAS1) vs. control HCT116 cells identified oxidative phosphorylation (OXPHOS) as the top enriched pathway. Dot size reflects gene count; color indicates statistical significance (−log_10_(*p*-value)). (**B**) JC-1 fluorescence microscopy to assess mitochondrial membrane potential (ΔΨm). Representative images of HCT116 cells from the indicated groups (NC, Sh-DIRAS1, Sh-DIRAS1 + OE-PHB1) after JC-1 staining. Red: J-aggregates (intact ΔΨm); green: JC-1 monomers (depolarized ΔΨm). Knockdown of DIRAS1 increased the green signal, indicating ΔΨm loss, which was partially rescued by PHB1 overexpression. Scale bar: 112.5 µm. (**C**) Flow cytometric analysis of JC-1 staining. Quantification of mitochondrial membrane potential by JC-1 dual-emission ratio in different treatment groups. (**D**) Quantification of JC-1 monomer signal by fluorescence microscopy. Green fluorescence intensity was measured from (**B**) using ImageJ V1.8.0.112. Data represent mean ± SEM (*n* = 9 fields/group from 3 experiments). One-way ANOVA with Tukey’s post hoc test: *** *p* < 0.001. (**E**) Quantification of JC-1 monomer-positive cells by flow cytometry. Percentage of cells with low ΔΨm (green fluorescence). Data: mean ± SEM (*n* = 3). Statistical analysis by unpaired two-tailed Student’s *t*-test: *** *p* < 0.001. (**F**) Assessment of mitochondrial permeability transition pore (mPTP) opening. Representative images of Calcein-AM-labeled HCT116 cells treated with cobalt chloride (quenching cytosolic fluorescence). Increased green fluorescence in the Sh-DIRAS1 group indicates enhanced mPTP opening, which was mitigated by OE-PHB1. Scale bar: 56.3 µm. (**G**) Measurement of mitochondrial reactive oxygen species (mtROS). Representative images showing MitoSOX Red fluorescence (mtROS) in indicated groups. DAPI: nuclear staining. Sh-DIRAS1 cells exhibited increased mtROS, reduced by PHB1 overexpression. Scale bar: 56.3 µm. (**H**) Quantification of mtROS intensity. MitoSOX Red signal from panel (**G**) was analyzed using ImageJ V1.8.0.112. Data are presented as mean ± SEM (*n* = 9 fields/group from 3 experiments). One-way ANOVA, Tukey’s test: *** *p* < 0.001. (**I**) Quantification of Calcein-AM fluorescence. Green fluorescence intensity (inversely reflecting mPTP opening) from panel (**F**) was quantified using ImageJ V1.8.0.112. Data: mean ± SEM (*n* = 9 fields/group from 3 experiments). One-way ANOVA, Tukey’s test: ** *p* < 0.01.

**Table 1 biology-14-00819-t001:** The primers used for plasmid constructs.

Gene	Sequences
DIRAS1-EGFP-F	CGAGCTCAAGCTTCGAATTCTATGCCGGAACAGAGTAACGA
DIRAS1-EGFP-R	GGGCGGGATCCGCGGCCGCTTAAGCGTAGTCTGGGACGTCGTATGGGTACATGAGGGTGCATTTGCC
DIRAS1-Sh-F	CCGGCCACAAATGTAGCAACCAGAACTCGAGTTCTGGTTGCTACATTTGTGGTTTTTG
DIRAS1-Sh-R	AATTCAAAAACCACAAATGTAGCAACCAGAACTCGAGTTCTGGTTGCTACATTTGTGG
PHB1-OE-F	CTAGCTAGCGCCACCATGGCTGCCAAAGTGTTTGA
PHB1-OE-R	CCGCTCGAGTTAAGCGTAGTCTGGGACGTCGTATGGGTACTGGGGCAGCTGGAGGAGCA
PHB1-Sh-F	CCGGTGTCATCTTTGACCGATTCCGCTCGAGCGGAATCGGTCAAAGATGACATTTTTG
PHB1-Sh-R	AATTCAAAAATGTCATCTTTGACCGATTCCGCTCGAGCGGAATCGGTCAAAGATGACA

## Data Availability

All relevant data supporting the findings of this study are included within the manuscript. Additional materials, data, and protocols are available from the corresponding authors upon reasonable request.

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
