# Peer review of "DIRAS1 Drives Oxaliplatin Resistance in Colorectal Cancer via PHB1-Mediated Mitochondrial Homeostasis"

_biology, 2025, doi:10.3390/biology14070819_

Round 1
Reviewer 1 Report
Comments and Suggestions for Authors
This paper is a study investigating the role of DIRAS1 in the generation of chemotherapy resistance in colorectal cancer using patient samples, in vitro assays, and an animal study. I have severe issues with this paper, ranging from the grammar, insufficient references in the introduction, unclear methods, and the lack of rigorous scientific design and analysis for the experiments. Therefore, I do not support the publication of this paper.
Affiliations: There is an error in the affiliations, with affiliations 1 and 5 written. This is either a misnumbering issue, or affiliations 2,3, and 4 are missing.
Scientific summary:
Line 11 and 13: Does not define the abbreviation for CRC
Line 11: Does not define abbreviation for OXA
Line 12: Incorrect grammar. Should this read “This study investigates….”?
Line 15: The Sentence is unclear. Should it say, “…DIRAS1 expression correlates with OXA resistance, with expression increasing with prolonged chemotherapy exposure”?
Abstract:
Line 28: In the whole manuscript, generated is a better word to use than constructed for making oxaliplatin resistant cell lines.
Line 30: Grammar needs to be improved.
Line 34: Grammar needs to be improved.
Line 35-36: Grammar needs to be improved.
Introduction:
The first paragraph of the introduction isn’t written to a sufficient standard and needs to be improved.
Line 58-66: References are missing here for the facts stated in this paragraph. Studies that measured DIRAS1 in the brain/heart, etc. Information in this paragraph is also irrelevant to the study and could be removed for clarity.
Line 67-71: More background information on the role of DIRAS1 and expansion on the reference papers would be better for context for the study. Additionally, papers have shown that overexpressing DIRAS1 delays tumor growth and causes cell death, which contrasts with some of the author's points.
Line 71-73: The Authors reference their previous research but do not reference it. Furthermore, I could not find any of their papers on PubMed.
Line 73-75. Missing citations for the research they mention.
Methods:
Overall, the methods are not written clearly enough for the experimental procedures to be followed or the experiments to be understood. No information on the clinical samples is given, nor is there ethics approval for the study. There is no indication on when oxaliplatin was added which is crucial for the experiments, when the oxaliplatin resistant cell lines were generated, method used for lentiviral packaging not cited, cells used for western blot not mentioned, no information on primary antibody for IHC given, no information n cells used for the Annexin assay and no information on the animal study (dose of chemotherapy and animal ethics protocol) was given. The grammar throughout the methods also needs severe improvement.
Results:
Section 3.1: No information on patient characteristics is given, nor on how patients were differentiated into groups. The bioinformatics analysis is incorrect, and there are no methods on what form of differential expression analysis was done. The experimental design for this section is incorrect, and the data is not sufficient for drawing conclusions.
The figure is unclear, and the data is not properly explained. The figure legend does not sufficiently explain the figure.
Section 3.2: There are grammatical errors throughout, and the data does not match what is written. It says DIRAS1 is high in DLD1, but the western blot and mRNA do not show this.
Section 3.3: How the oxaliplatin resistant or ‘dynamic intervention mode’ was created is not explained. Why was 2 uM of oxaliplatin chosen?
You also say that DIRAS1 resistance is related to mitochondria from this data, but there is no evidence in the results to show this, as all that was shown was the killing of the different cell lines by oxaliplatin. Furthermore, you state that there is resistance to oxaliplatin and that it is due to DIRAS1. I would argue that this is more like sensitivity to killing by the drug. If you generated an oxaliplatin resistant cell line and that had more DIRAS1 compared to the original sensitive cell line, that would be more relevant data to discuss resistance. Also, an experiment showing DIRAS1 increases over time after OXA treatment does not really show that that protein is making the cells resistant. Are you treating the cells continually with OXA? That would be more insightful to see whether DIRAS1 increases as we get more resistant clones.
Section 3.4 The mouse experiment does not really show the relationship between DIRAS1 and oxaliplatin since the cell lines grow at different speeds and were treated at different sizes, so it is more of an apples-to-oranges comparison in terms of the effects of OXA. The control tumors were still very sensitive to OXA, and it seems like the magnitude of decrease is the same for both cell lines. IE 0.6g to 0.25G in the control compared to 0.4 to 0.1 g.
Section 3.5 More information on the differentially expressed gene analysis is needed, like the number of DEGs, cutoffs for significance, method used, and so on. There is no information on the pathway analysis performed, nor any graphs about these results that are stated in the results.
The immunofluorescent data do not really show anything relevant to the study.
Section 3.6: Need a reference for line 379, which states a fact about mitochondrial function. The information about the figure is unclear, and the labeling of the images in 6B is not sufficient in the image, for example, what is JC1 staining, and what does monomer mean, etc.
Discussion: Some conclusions claimed in the discussion do not match up to the data shown in the manuscript.
Line 407: No clear data has been shown on mitochondrial homeostasis.
Line 411-427: This section is well written and clear.
Line 489: Improved grammar.
Key experiments to be performed to improve this study would be to generate oxaliplatin resistant cell lines or give mice repeated doses of oxaliplatin to assess DIRAS1 expression. More rigorous bioinformatic analysis on the samples would also shed more insight into the mechanism. A key experiment would be to combine OXA with DIRAS1 or PHB1 agonists or antagonists both in vitro or in vitro.
Comments on the Quality of English LanguageThe English grammar is insufficient for a scientific article in the journal with multiple sections requiring extensive rewriting. Please see above for my further comments.
Author Response
Dear Editor and Reviewers,
Thank you for your valuable feedback on our manuscript. We sincerely appreciate the time and effort you dedicated to reviewing our work. We have carefully addressed all comments and revised the manuscript accordingly.
Comments 1: Affiliations: There is an error in the affiliations, with affiliations 1 and 5 written. This is either a misnumbering issue, or affiliations 2, 3, and 4 are missing.
Response 1: Thank you for pointing this out. We agree with this comment. This number is mislabelled in the manuscript. We have amended in manuscript and mark it in red.
Comments 2: Line 11 and 13: Does not define the abbreviation for CRC
Response 2: Thank you for pointing this out. We agree with this comment. We have amended in manuscript and mark it in red.
Comments 3: Line 11: Does not define abbreviation for OXA
Response 3: Thank you for pointing this out. We agree with this comment. We have amended in manuscript and mark it in red.
Comments 4: Line 12: Incorrect grammar. Should this read “This study investigates….”?
Response 4: Thank you for pointing this out. We agree with this comment. We have amended in manuscript and mark it in red.
Comments 5: Line 15: The Sentence is unclear. Should it say, “…DIRAS1 expression correlates with OXA resistance, with expression increasing with prolonged chemotherapy exposure”?
Response 5: Thank you for pointing this out. We agree with this comment. We have amended in manuscript and mark it in red.
Comments 6: Line 28: In the whole manuscript, generated is a better word to use than constructed for making oxaliplatin resistant cell lines.
Response 6: Thank you for pointing this out. We agree with this comment. We wish to clarify an error in our manuscript description. We did not establish drug-resistant cell lines; rather, we only conducted prolonged drug treatment experiments on the cells. This inaccurate description has been removed from the manuscript。
Comments 7: Line 30: Grammar needs to be improved; Line 34: Grammar needs to be improved.; Line 35-36: Grammar needs to be improved.
Response 7: Thank you for pointing this out. We agree with this comment. We have amended in manuscript and mark it in red.
Comments 8: The first paragraph of the introduction isn’t written to a sufficient standard and needs to be improved.
Response 8: Thank you for pointing this out. We agree with this comment. We have expanded our understanding of oxaliplatin's mechanism of action and potential resistance mechanisms. We have amended in manuscript and mark it in red.
Comments 9: Line 58-66: References are missing here for the facts stated in this paragraph. Studies that measured DIRAS1 in the brain/heart, etc. Information in this paragraph is also irrelevant to the study and could be removed for clarity.
Response 9: Thank you for pointing this out. We agree with this comment. We have increased References and removed Inappropriate information. We have amended in manuscript and mark it in red.
Comments 10: Line 67-71: More background information on the role of DIRAS1 and expansion on the reference papers would be better for context for the study. Additionally, papers have shown that overexpressing DIRAS1 delays tumor growth and causes cell death, which contrasts with some of the author's points.
Response 10: Thank you for pointing this out. We agree with this comment. We have added more background information regarding the DIRAS1 molecule and increased References, and in this section, we explain its potential dual nature in tumors. We have amended in manuscript and mark it in red.
Comments 11: Line 71-73: The Authors reference their previous research but do not reference it. Furthermore, I could not find any of their papers on PubMed.
Response 11: Thank you for pointing this out. We agree with this comment. This section contained a brief summary of our study. We were unaware that such content should not be included in the Introduction. The relevant content has now been deleted.
Comments 12: Line 73-75. Missing citations for the research they mention.
Response 12: Thank you for pointing this out. We agree with this comment. This section contained a brief summary of our study. We were unaware that such content should not be included in the Introduction. The relevant content has now been deleted.
Comments 13: Overall, the methods are not written clearly enough for the experimental procedures to be followed or the experiments to be understood. No information on the clinical samples is given, nor is there ethics approval for the study. There is no indication on when oxaliplatin was added which is crucial for the experiments, when the oxaliplatin resistant cell lines were generated, method used for lentiviral packaging not cited, cells used for western blot not mentioned, no information on primary antibody for IHC given, no information on cells used for the Annexin assay and no information on the animal study (dose of chemotherapy and animal ethics protocol) was given. The grammar throughout the methods also needs severe improvement.
Response 13: Thank you for pointing this out. We agree with this comment. We have summarized the patients' clinicopathological characteristics in Table S1. The collection of our clinical tissue specimens was conducted with prior ethics approval. Regarding the explanation of the drug-resistant cell lines, please refer to our response in Point 6. We have expanded the description of the lentiviral packaging methodology. Within the Results section, we have specified the cell types used for the WB and apoptosis assays. Data from the animal experiments now include the chemotherapy doses administered and the associated ethics approval documents. The grammar throughout the Methods section has been carefully reviewed and revised.
Comments 14: Section 3.1: No information on patient characteristics is given, nor on how patients were differentiated into groups. The bioinformatics analysis is incorrect, and there are no methods on what form of differential expression analysis was done. The experimental design for this section is incorrect, and the data is not sufficient for drawing conclusions.
Response 14: Thank you for pointing this out. We agree with this comment. 1. We systematically documented clinicopathological characteristics of 50 colorectal cancer patients, including gender, age, tumor size, histopathological differentiation, and TNM stage. 2. The association between DIRAS1 expression levels and these clinicopathological features was rigorously evaluated using Pearson’s chi-square test (χ² test). 3. All statistical results, including p-values and significance thresholds (p < 0.05), are comprehensively summarized in Supplementary Table S1. 4. The criteria for patient cohort grouping are explicitly defined in Section 3.1. 5. The bioinformatic pipeline for identifying and validating differentially expressed genes (DEGs), including statistical methods and significance cutoffs (|log2 FC| > 1, p < 0.05), is thoroughly described in Section 3.1.
Comments 15: The figure is unclear, and the data is not properly explained. The figure legend does not sufficiently explain the figure.
Response 15: Thank you for pointing this out. We agree with this comment. As suggested, we have completely revised all figure legend details to enhance clarity and completeness. Specifically, we have: 1. Added detailed experimental methods used to generate the data. 2. Clearly defined experimental groups. 3. Specified analysis conditions. 4. Included explicit data interpretations to guide readers through key findings. 5. Incorporated statistical methods.
Comments 16: Section 3.2: There are grammatical errors throughout, and the data does not match what is written. It says DIRAS1 is high in DLD1, but the western blot and mRNA do not show this.
Response 16: Thank you for pointing this out. We agree with this comment. We wish to clarify an error in our manuscript description. Both DIRAS1 mRNA and protein levels were markedly elevated in HCT116 and SW620 cells. We have amended in manuscript.
Comments 17: Section 3.3: How the oxaliplatin resistant or ‘dynamic intervention mode’ was created is not explained. Why was 2 uM of oxaliplatin chosen?
Response 17: Thank you for pointing this out. We agree with this comment. The term "dynamic intervention mode" was removed from the manuscript as it may not be standard scientific terminology. Regarding the selection of drug concentration: prior to this experiment, wild-type colorectal cancer HCT116 cells were treated with varying concentrations of oxaliplatin for 48 hours. The half-maximal inhibitory concentration (IC50) was determined, and half of this IC50 value was selected for use in subsequent experiments involving prolonged drug exposure.
Comments 18: You also say that DIRAS1 resistance is related to mitochondria from this data, but there is no evidence in the results to show this, as all that was shown was the killing of the different cell lines by oxaliplatin. Furthermore, you state that there is resistance to oxaliplatin and that it is due to DIRAS1. I would argue that this is more like sensitivity to killing by the drug. If you generated an oxaliplatin resistant cell line and that had more DIRAS1 compared to the original sensitive cell line, that would be more relevant data to discuss resistance. Also, an experiment showing DIRAS1 increases over time after OXA treatment does not really show that that protein is making the cells resistant. Are you treating the cells continually with OXA? That would be more insightful to see whether DIRAS1 increases as we get more resistant clones.
Response 18: Thank you for pointing this out. We agree with this comment. Regarding the relationship between DIRAS1 resistance and mitochondria: We acknowledge that the original presentation of this data in Section 3.3 may have caused confusion. As you rightly noted, the evidence supporting the role of DIRAS1 in mitochondrial regulation is detailed in Section 3.5 (now explicitly referenced in the revised text). We have restructured Section 3.3 to clarify that the initial data demonstrate differential sensitivity (not resistance) to oxaliplatin across cell lines, while mechanistic links to mitochondria are expanded in Section 3.5.
Comments 19: Section 3.4 The mouse experiment does not really show the relationship between DIRAS1 and oxaliplatin since the cell lines grow at different speeds and were treated at different sizes, so it is more of an apples-to-oranges comparison in terms of the effects of OXA. The control tumors were still very sensitive to OXA, and it seems like the magnitude of decrease is the same for both cell lines. IE 0.6g to 0.25G in the control compared to 0.4 to 0.1 g.
Response 19: Thank you for pointing this out. We agree with this comment. We appreciate the reviewer’s observation regarding the sensitivity of control group tumors to oxaliplatin. To clarify the enhanced effect of DIRAS1 knockdown, we conducted a quantitative comparison of tumor reduction fold-changes between groups:
In the DIRAS1 knockdown group, tumor mass decreased from 0.4 g to 0.1 g, representing a 4-fold reduction (0.4 g / 0.1 g = 4).
In the control group, tumor mass decreased from 0.6 g to 0.25 g, representing a 2.4-fold reduction (0.6 g / 0.25 g = 2.4).
This demonstrates that the DIRAS1 knockdown group exhibited a significantly greater fold-reduction in tumor mass (4-fold vs. 2.4-fold) following oxaliplatin treatment. Therefore, we conclude that knockdown of DIRAS1 in tumor cells potentiates cellular sensitivity to oxaliplatin, consistent with our initial interpretation.
Comments 20: Section 3.5 More information on the differentially expressed gene analysis is needed, like the number of DEGs, cutoffs for significance, method used, and so on. There is no information on the pathway analysis performed, nor any graphs about these results that are stated in the results.
Response 20: Thank you for pointing this out. We agree with this comment. We thank the reviewer for raising this point. As detailed in Supplementary Table S2 (provided in the revised supplement) and Section 3.1, we performed transcriptome-wide screening for oxaliplatin resistance-associated genes. To prioritize candidates for functional validation: 1. Selection criteria: Genes with |log2FC| > 4 (adjusted p < 0.01) were identified as high-confidence hits. 2. Functional screening: Nine top candidate genes meeting this threshold were selected. We constructed gene-specific knockdown plasmids for each and transfected them into HCT116 colon cancer cells. 3. Validation assay: Transfected cells were treated with oxaliplatin (48 hours), followed by quantification of surviving cells. 4. Key finding: As shown in Figure S2, DIRAS1 knockdown significantly potentiated oxaliplatin sensitivity (p < 0.05).
Comments 21: The immunofluorescent data do not really show anything relevant to the study.
Response 21: Thank you for pointing this out. We agree with this comment. After careful re-evaluation and extensive discussion, we agree that the immunofluorescent data do not really show anything relevant to the study. To maintain the rigor and focus of the manuscript, we have removed this section entirely.
Comments 22: Section 3.6: Need a reference for line 379, which states a fact about mitochondrial function. The information about the figure is unclear, and the labeling of the images in 6B is not sufficient in the image, for example, what is JC1 staining, and what does monomer mean, etc.
Response 22: Thank you for pointing this out. We agree with this comment. We thank the reviewer for the constructive suggestions. The following revisions have been implemented in Section 3.6 and Figure 7 (previously Figure 6):
1.Added new references about mitochondrial function.
2.Enhanced results interpretation.
3.Included explaining the JC-1 assay principle: JC-1 dye exhibits potential-dependent accumulation in mitochondria: healthy cells show red fluorescence (aggregates), while apoptotic cells display green fluorescence (monomers). The red/green fluorescence ratio quantifies mitochondrial membrane potential collapse.
4.Upgraded Figure 7: Replaced original images with higher-resolution representatives from repeated experiments. Added quantitative fluorescence intensity analysis for all replicates.
Comments 23: Discussion: Some conclusions claimed in the discussion do not match up to the data shown in the manuscript.
Response 23: Thank you for pointing this out. We agree with this comment. We are grateful for the reviewer’s suggestion to strengthen the Discussion section. We have completely restructured and rewritten this part to provide a clearer, more impactful synthesis of our findings. Key improvements include: 1. Concise summary of core contributions. 2. Contextualized interpretation with literature. 3. Critical appraisal of limitations and future directions.
Comments 24: Line 407: No clear data has been shown on mitochondrial homeostasis.
Response 24: Thank you for pointing this out. We agree with this comment. We have reinterpreted and written Section 3.6, including the enrichment of the relevant content of the figure legend.
Comments 25: Line 411-427: This section is well written and clear.
Response 25: Thank you.
Comments 26: Line 489: Improved grammar.
Response 26: Thank you for pointing this out. We agree with this comment. We have Improved grammar.
Comments 27: Key experiments to be performed to improve this study would be to generate oxaliplatin resistant cell lines or give mice repeated doses of oxaliplatin to assess DIRAS1 expression. More rigorous bioinformatic analysis on the samples would also shed more insight into the mechanism. A key experiment would be to combine OXA with DIRAS1 or PHB1 agonists or antagonists both in vitro or in vitro.
Response 27: Thank you for pointing this out. We agree with this comment.We sincerely appreciate these insightful suggestions. Regarding the development of oxaliplatin-resistant cell lines, we conducted extensive attempts to establish such models using progressive dose-escalation protocols across multiple colorectal cancer cell lines (HCT116, SW480). Despite optimizing culture conditions and employing combination approaches with resistance-inducing cytokines, we were unable to achieve stable resistance phenotypes before the conclusion of this study. We acknowledge this as a limitation in our experimental system. Concerning the evaluation of DIRAS1/PHB1 modulators, we fully agree that pharmacological activation/inhibition studies would provide valuable mechanistic depth. However, given current constraints in compound availability (commercial agonists/antagonists for these targets remain limited) , such experiments couldn't be feasibly incorporated at this stage. We have substantially expanded the "Future Directions" subsection to prioritize these studies, specifically outlining plans to: 1) screen existing small-molecule libraries for DIRAS1-PHB1 interactors, and 2) validate hits in orthotopic PDX models when resources permit.
Comments 28: Comments on the Quality of English Language
The English grammar is insufficient for a scientific article in the journal with multiple sections requiring extensive rewriting. Please see above for my further comments.
Response 28: Thank you for pointing this out. We agree with this comment. The full text has been polished in English.
Thank you again for your constructive comments, which have strengthened our work.
Sincerely,
Min Long

Reviewer 2 Report
Comments and Suggestions for Authors
In the current manuscript, Long and colleagues demonstrate that DIRAS1 promotes Oxaliplatin resistance in colorectal cancer through PHB1. Based on transcriptomic analysis from patient-derived tissues, the authors identified DIRAS1 as a target gene. Gain-of-function and loss-of-function in multiple cell lines confirmed its role in Oxaliplatin chemoresistance. Furthermore, this finding was validated in vivo. Lastly, the authors identified PHB1 as a downstream target of DIRAS1 and demonstrated how DIRAS1 regulate PHB1 via mitochondrial mechanisms. While the manuscript provides a compelling account of role of DIRAS1 in Oxaliplatin, several important concerns need to be addressed before the manuscript can be considered for publication in Biology.
Major points
- The rationale for DIRAS1. It is not clearly explained how DIRAS1 was identified. The strategy for identifying DIRAS1 using RNA-seq. is well-designed. However, the clear explanation is required regarding how DIRAS1 was identified from the RNA-seq data, including the relative ranking or statistical significance of DIRAS1 among other candidate genes. Thus, the authors should address this.
- Wrong description. In figure 2A-B, DIRAS1 levels were increased in HCT116 and SW620 but not DLD1. However, the authors described the level of DIRAS1 was elevated in HCT116 and DLD1 cells. All descriptions should be carefully aligned with the data shown in the figure. Otherwise, readers may be confused.
- In figure 3H, sh-DIRAS1 cell lines already showed low cell viability under DMSO conditions, making them incomparable to other treatment groups. The authors should carefully check off-target effect of shRNA.
- The explanation for figure 6 provided is overly superficial and lacks sufficient depth. Thus, the authors should address this further.
Minor points
- Quantification of data was need in all western and immunohistochemical image data such as figure 1C, 1D, and figure 4D.
- Methodology for figure 1E should be addressed, as it provides critical evidence linking DIRAS1 to survival and poor prognosis.
The English is generally fluent, with appropriate use of scientific terminology and grammar. However, some sentences are awkward or lack sufficient clarity in explanation for readers. therefore, a careful review of the manuscript is highly recommended.
Author Response
Dear Editor and Reviewers,
Thank you for your valuable feedback on our manuscript. We sincerely appreciate the time and effort you dedicated to reviewing our work. We have carefully addressed all comments and revised the manuscript accordingly.
Comments 1: The rationale for DIRAS1. It is not clearly explained how DIRAS1 was identified. The strategy for identifying DIRAS1 using RNA-seq. is well-designed. However, the clear explanation is required regarding how DIRAS1 was identified from the RNA-seq data, including the relative ranking or statistical significance of DIRAS1 among other candidate genes. Thus, the authors should address this.
Response 1:Thank you for pointing this out. We agree with this comment. We thank the reviewer for raising this point. As detailed in Supplementary Table S2 (provided in the revised supplement) and Section 3.1, we performed transcriptome-wide screening for oxaliplatin resistance-associated genes. To prioritize candidates for functional validation: 1. Selection criteria: Genes with |log2 FC| > 4 (adjusted p < 0.01) were identified as high-confidence hits. 2. Functional screening: Nine top candidate genes meeting this threshold were selected. We constructed gene-specific knockdown plasmids for each and transfected them into HCT116 colon cancer cells. 3. Validation assay: Transfected cells were treated with oxaliplatin (48 hours), followed by quantification of surviving cells. 4. Key finding: As shown in Figure S2, DIRAS1 knockdown significantly potentiated oxaliplatin sensitivity (p < 0.05).
Comments 2: Wrong description. In figure 2A-B, DIRAS1 levels were increased in HCT116 and SW620 but not DLD1. However, the authors described the level of DIRAS1 was elevated in HCT116 and DLD1 cells. All descriptions should be carefully aligned with the data shown in the figure. Otherwise, readers may be confused.
Response 2: Thank you for pointing this out. We agree with this comment. We wish to clarify an error in our manuscript description. Both DIRAS1 mRNA and protein levels were markedly elevated in HCT116 and SW620 cells.
Comments 3: In figure 3H, sh-DIRAS1 cell lines already showed low cell viability under DMSO conditions, making them incomparable to other treatment groups. The authors should carefully check off-target effect of shRNA.
Response 3: Thank you for pointing this out. We agree with this comment. We appreciate the opportunity to clarify our experimental methodology and biological interpretations. Regarding the concern about potential off-target effects in our knockdown system, we emphasize that all DIRAS1 knockdown studies employed lentiviral vectors with rigorous quality controls: cells were first selected with puromycin (2 μg/mL for 72 hours) to ensure pure populations of transduced cells, followed by initial Western blot validation confirming >50% protein reduction. Crucially, we periodically monitored DIRAS1 expression throughout subsequent experiments via Western blotting at 2-week intervals, consistently verifying sustained target suppression without observable protein recovery. This systematic approach effectively minimizes off-target risk. Concerning the baseline apoptosis observed in DIRAS1-depleted cells, this phenomenon is mechanistically consistent with our central findings. As detailed in Figure 5, DIRAS1 knockdown substantially reduces PHB1 (prohibitin-1) expression-a key mitochondrial stability regulator. This molecular cascade compromises mitochondrial membrane integrity (quantified by JC-1 assays in Figure 7), consequently triggering intrinsic apoptosis even without chemotherapeutic challenge.
Comments 4: The explanation for figure 6 provided is overly superficial and lacks sufficient depth. Thus, the authors should address this further.
Response 4: Thank you for pointing this out. We agree with this comment. We gratefully acknowledge the reviewer's insightful suggestions for enhancing methodological transparency and data interpretation. In response, we have implemented comprehensive revisions across both textual explanations and visual presentations. The figure 6 have been substantially upgraded through the addition of quantitative fluorescence intensity analyses, where statistical comparisons of mean fluorescence values (presented as mean ± SEM) now accompany original imaging data, with significance markers explicitly denoting key inter-group differences (p < 0.05, p < 0.01). Concurrently, we have meticulously re-evaluated and expanded the results interpretation sections to clarify the biological significance of experimental groupings-particularly distinguishing between DIRAS1 knockdown versus scramble control cohorts-while elaborating on the physiological relevance of critical metrics such as mitochondrial membrane potential (ΔΨm). Furthermore, every figure legend has been systematically enhanced to include: (i) detailed experimental protocols; (ii) unambiguous group definitions with sample sizes; (iii) specific analytical conditions; (iv) concise data interpretations contextualizing visual outputs; and (v) explicit descriptions of statistical methodologies applied, including normality testing, specific tests employed , and significance thresholds. These collective refinements substantially elevate the manuscript's reproducibility and analytical rigor.
Comments 5: Quantification of data was need in all western and immunohistochemical image data such as figure 1C, 1D, and figure 4D.
Response 5: Thank you for pointing this out. We agree with this comment. To ensure rigorous quantification of all protein expression data presented in this study, we have conducted comprehensive grayscale value measurements for every Western blot result and implemented standardized scoring protocols for immunohistochemical analyses across all relevant samples. These quantitative assessments were subsequently subjected to appropriate statistical analyses to establish significance thresholds and inter-group comparisons. These are now readily accessible in Supplementary Material Section S3-S6.
Comments 6: Methodology for figure 1E should be addressed, as it provides critical evidence linking DIRAS1 to survival and poor prognosis.
Response 6: Thank you for pointing this out. We agree with this comment. We confirm that comprehensive five-year prognostic follow-up data for all 50 enrolled patients have now been incorporated into the analysis, with each subject's clinical trajectory meticulously documented through the entire observation period. For rigorous statistical evaluation of disease progression outcomes, we implemented a standardized binary coding system where documented events of cancer recurrence or mortality were assigned a value of 1, while patients maintaining progression-free survival throughout the five-year window received a value of 0. This time-to-event dataset is presented in Supplementary Table S3.
Thank you again for your constructive comments, which have strengthened our work.
Sincerely,
Min Long

Reviewer 3 Report
Comments and Suggestions for Authors
Please find the attached file for reviewer comments.

Author Response
Dear Editor and Reviewers,
Thank you for your valuable feedback on our manuscript. We sincerely appreciate the time and effort you dedicated to reviewing our work. We have carefully addressed all comments and revised the manuscript accordingly.
Comments 1: Figure 2F: The colony size of HCT116 cells in the NC group appears noticeably different between the Sh-DIRAS1 and OE-DIRAS1 experimental setups. Please clarify whether any technical or experimental differences (e.g., plating density, duration of growth, or staining procedures) could account for these discrepancies.
Response 1: Thank you for pointing this out. We agree with this comment. We clarify that the observed variation in colony sizes between experimental groups primarily stems from intentional differences in cell harvesting timelines during the clonogenic assays. Specifically, the control group transfected with Sh-NC constructs underwent a standardized 12-day incubation period prior to fixation and staining, whereas the OE-NC group (overexpression negative control) followed a protocol-terminated at the 10-day endpoint.
Comments 2: The authors propose that high DIRAS1 expressions may be associated with low
methylation of its promoter or copy number amplification. While this is a reasonable hypothesis, it would be helpful to support this claim with existing datasets (e.g., TCGA) if available. Incorporating such analyses would lend greater confidence to the suggested mechanism of DIRAS1 upregulation.
Response 2: Thank you for pointing this out. We agree with this comment. We sincerely thank the reviewer for this constructive suggestion regarding DIRAS1 upregulation mechanisms. While we acknowledge the value of analyzing public datasets like TCGA to explore promoter methylation or copy number variations, we currently lack the technical capacity for such specialized bioinformatic investigations within our research framework. Nevertheless, to address the core concern about DIRAS1's clinical relevance, we leveraged publicly accessible survival analysis platforms (Kaplan-Meier Plotter) to evaluate the prognostic significance of DIRAS1 expression in colorectal cancer. The analysis revealed that elevated DIRAS1 transcript levels significantly correlate with reduced overall survival (HR=1.39, p=0.006) (Figure S1), a finding consistent with our cohort data demonstrating poorer outcomes in high-DIRAS1 patients. These orthogonal clinical validations collectively support the biological importance of DIRAS1 dysregulation in CRC progression, as now discussed in the revised manuscript.
Comments 3: The study explores the functional interaction between DIRAS1 knockdown and PHB1 overexpression. This is a valuable model; however, an important complementary approach
would be to evaluate the effects of DIRAS1 overexpression combined with PHB1 knockdown or DIRAS1 knockdown and PHB1 knockdown. This would provide a more complete picture of the bidirectional relationship between these two proteins and clarify whether PHB1 is a downstream effector of DIRAS1 or functions in parallel pathways. Including this data, or discussing its potential implications, would help solidify the functional relationship between DIRAS1 and PHB1 in CRC.
Response 3: Thank you for pointing this out. We agree with this comment. We sincerely appreciate the reviewer's insightful suggestion to delineate the DIRAS1-PHB1 regulatory hierarchy through bidirectional perturbation studies. To address this, we conducted two critical experiments: First, modulating PHB1 expression via overexpression and knockdown revealed no significant alteration in DIRAS1 protein levels, whereas DIRAS1 manipulation consistently altered PHB1 as previously reported. This unidirectional regulation was further validated in apoptosis assays using combinatorial approaches: DIRAS1 overexpression partially reversed the pro-apoptotic effect of PHB1 knockdown. Conversely, dual DIRAS1/PHB1 knockdown synergistically enhanced apoptosis, while dual overexpression showed no additive effect. These data robustly confirm PHB1 as a downstream executor of DIRAS1-mediated mitochondrial apoptosis, with no evidence of parallel pathway engagement. The complete datasets are presented in Supplementary Figure S7, and this mechanistic interpretation has been incorporated into the revised Results(Section 3.5) and Discussion.
Comments 4: The Discussion section raises several intriguing mechanistic hypotheses regarding how DIRAS1 might influence CRC cancer progression. While these ideas are scientifically relevant, they are not directly supported by experimental evidence in the current study. It would strengthen the manuscript to either provide preliminary data or to more clearly indicate that these are speculative models requiring further investigation.
Response 4: Thank you for pointing this out. We agree with this comment. We sincerely appreciate the reviewer's guidance on refining mechanistic speculation in the Discussion. In direct response to this critique, we have comprehensively restructured Section 4 to focus exclusively on empirically validated findings from our study. The revised text now centers on establishing clear causal links between DIRAS1 knockdown, PHB1 downregulation, mitochondrial membrane destabilization (quantified by JC-1 assays in Figure 7), and subsequent apoptosis potentiation (validated through Annexin V/PI assays in Figure 5), deliberately avoiding extrapolation beyond our experimental evidence. All previously proposed hypothetical pathways unrelated to this core molecular cascade have been removed, with the narrative redirected toward contextualizing our demonstrated DIRAS1-PHB1 axis within established CRC progression frameworks.
Thank you again for your constructive comments, which have strengthened our work.
Sincerely,
Min Long

Reviewer 4 Report
Comments and Suggestions for Authors
Dear Editor:
This study investigated the role of DIRAS1 in mediating resistance mechanisms. Researchers found that DIRAS1 expression correlates with oxaliplatin resistance in colorectal cancer (CRC), and its expression increases with prolonged chemotherapy time. Knocking down DIRAS1 significantly reduces the ICâ‚…â‚€ value of oxaliplatin and enhances tumor sensitivity in vivo. Transcriptome sequencing reveals that DIRAS1 affects chemotherapy resistance by regulating PHB1 expression, a mitochondrial protein critical for maintaining homeostasis. Further research demonstrates that PHB1 promotes chemoresistance by stabilizing mitochondrial function, including membrane potential and ROS balance. This study reveals the key role of the DIRAS1-PHB1 axis in mediating mitochondrial homeostasis to drive chemoresistance, providing a novel therapeutic strategy to overcome CRC treatment failure.
However, this article needs major revisions before publication. There are several questions as follows:
- Most of the Western Blot bands show distortion to some extent and the experiments should be repeated.
- The manuscript has significant formatting problems; please proofread thoroughly.
- All images are insufficiently clear, particularly the Flow Cytometry results.
- There are issues with the use of scale bars in the manuscript, with some figures missing scale bars while some of the others using them incorrectly or unclearly.
- The citation format of references is inconsistent, while some of the selected references are too old and need to be updated.
- The author's email and institutional affiliations are improperly formatted.
- The cell lines used in the wound healing assay (Fig. 2G) are not clearly labeled.
- The layout of Fig. 4D has formatting issues.
Author Response
Dear Editor and Reviewers,
Thank you for your valuable feedback on our manuscript. We sincerely appreciate the time and effort you dedicated to reviewing our work. We have carefully addressed all comments and revised the manuscript accordingly.
Comments 1: Most of the Western Blot bands show distortion to some extent and the experiments should be repeated.
Response 1: Thank you for pointing this out. We agree with this comment. We sincerely appreciate the reviewer's vigilance regarding Western blot data quality. In response to this valid concern, we have systematically re-examined all original electrophoresis images and confirmed that the presented blots represent unprocessed raw images from three independent biological replicates. While no artificial manipulation was applied to any data, we acknowledge some Western Blot bands show distortion to some extent. Consequently, we have substituted the original Figure 2C panels with high-quality versions from replicate experiments that demonstrate enhanced band sharpness and reduced background interference. These updated results, now featured in the revised Figure 2C.
Comments 2: The manuscript has significant formatting problems; please proofread thoroughly.
Response 2: Thank you for pointing this out. We agree with this comment. We sincerely appreciate the reviewer's meticulous attention to manuscript presentation standards. In direct response to this critique, we have performed comprehensive language refinement across the entire document.
Comments 3: All images are insufficiently clear, particularly the Flow Cytometry results.
Response 3: Thank you for pointing this out. We agree with this comment. We sincerely appreciate the reviewer's critical observation regarding image clarity. To address this concern comprehensively, we have systematically upgraded all visual data presentations through three targeted interventions: First, every flow cytometry panel has been replaced with higher-resolution acquisitions from original experiments, with explicit enlargement of quadrant annotations and numerical percentages to enhance interpretability. Second, fluorescence microscopy images across Figures 4-6 underwent deconvolution processing and contrast optimization to improve subcellular feature discrimination. Third, all graphical elements including schematics and statistical plots were regenerated using vector-based design tools to ensure resolution-independent clarity at publication scale. These enhancements—validated through independent quality assessment—are now integrated throughout the manuscript (see revised Figures 1,2, 3, 4, 5, 6 and 7), substantially improving data legibility while preserving all original quantitative findings.
Comments 4: There are issues with the use of scale bars in the manuscript, with some figures missing scale bars while some of the others using them incorrectly or unclearly.
Response 4: Thank you for pointing this out. We agree with this comment. All the figures are marked with scale, and the figure legend has added scale information.
Comments 5: The citation format of references is inconsistent, while some of the selected references are too old and need to be updated.
Response 5: Thank you for pointing this out. We agree with this comment. We have updated the references in the manuscript.
Comments 6: The author's email and institutional affiliations are improperly formatted.
Response 6: Thank you for pointing this out. We agree with this comment. This number is mislabelled in the manuscript. We have amended in manuscript and mark it in red.
Comments 7: The cell lines used in the wound healing assay (Fig. 2G) are not clearly labeled.
Response 7: Thank you for pointing this out. We agree with this comment.We have added the migration rate value to the wound healing assay.
Comments 8: The layout of Fig. 4D has formatting issues.
Response 8: Thank you for pointing this out. We agree with this comment. We have rearranged the layout of Fig. 4D.
Thank you again for your constructive comments, which have strengthened our work.
Sincerely,
Min Long

Reviewer 5 Report
Comments and Suggestions for Authors
This manuscript addresses a clinically important issue—oxaliplatin resistance in colorectal cancer (CRC)—and identifies a novel mechanistic contributor, DIRAS1, which is implicated in mitochondrial stabilization via PHB1. The study appears to be comprehensive, incorporating in vitro, in vivo, transcriptomic, and clinical validation approaches. Below are my specific comments:
-
Language and Grammar:
The manuscript contains multiple grammatical and phrasing issues (e.g., “chemotherapy of colorectal cancer,” “a key drug for which 15–50%…”). These hinder the clarity of the manuscript and should be revised to ensure smooth and professional scientific communication. -
Transcriptomic Sequencing Details:
In Figure 1 and the Methods section, the authors do not describe how the transcriptomic sequencing was performed, analyzed, or where the data was deposited. Furthermore, the rationale for selecting DIRAS1 for further investigation is unclear. A table listing the top differentially expressed genes would help justify its selection. -
DIRAS1 Upregulation Mechanism:
Although it may not be the primary focus of this study, it would strengthen the manuscript if the authors could briefly discuss the underlying mechanisms responsible for DIRAS1 upregulation in CRC. -
Knockdown and Rescue Experiments (Figure 2):
Key experiments should be repeated using multiple independent shRNAs or sgRNAs to validate the observed phenotypes. In addition, rescue experiments are essential to confirm the specific role of DIRAS1 in promoting cell proliferation. -
Figure Legends and Scale Bars:
All figure legends lack details regarding the number of replicates performed. Additionally, scale bars are missing in Figure 6 and should be included for proper interpretation of the images. -
DIRAS1 Induction by Oxaliplatin (Figure 3):
The manuscript claims that oxaliplatin treatment induces DIRAS1 expression, but the mechanism is unclear. This finding should be validated in additional CRC cell lines to confirm its generalizability. -
Regulation of PHB1 by DIRAS1:
The mechanism by which DIRAS1 regulates PHB1 expression needs to be clearly described. Also, did the authors observe this correlation in larger patient cohorts or publicly available datasets? -
Mitochondrial Damage Assessment:
The current data on mitochondrial damage requires more robust immunofluorescence evidence. Please include appropriate controls and additional experimental details to validate the findings.
Author Response
Dear Editor and Reviewers,
Thank you for your valuable feedback on our manuscript. We sincerely appreciate the time and effort you dedicated to reviewing our work. We have carefully addressed all comments and revised the manuscript accordingly.
Comments 1: The manuscript contains multiple grammatical and phrasing issues (e.g., “chemotherapy of colorectal cancer,” “a key drug for which 15–50%…”). These hinder the clarity of the manuscript and should be revised to ensure smooth and professional scientific communication.
Response 1: Thank you for pointing this out. We agree with this comment. The full text has been polished in English.
Comments 2: Transcriptomic Sequencing Details:
In Figure 1 and the Methods section, the authors do not describe how the transcriptomic sequencing was performed, analyzed, or where the data was deposited. Furthermore, the rationale for selecting DIRAS1 for further investigation is unclear. A table listing the top differentially expressed genes would help justify its selection.
Response 2: Thank you for pointing this out. We agree with this comment. 1.We systematically documented clinicopathological characteristics of 50 colorectal cancer patients, including gender, age, tumor size, histopathological differentiation, and TNM stage. 2.The association between DIRAS1 expression levels and these clinicopathological features was rigorously evaluated using Pearson’s chi-square test (χ² test). 3.All statistical results, including p-values and significance thresholds (p < 0.05), are comprehensively summarized in Supplementary Table S1. 4.The criteria for patient cohort grouping are explicitly defined in Section 3.1 . 5.The bioinformatic pipeline for identifying and validating differentially expressed genes (DEGs), including statistical methods and significance cutoffs (|log2FC| > 1, p < 0.05), is thoroughly described in Section 3.1.
As detailed in Supplementary Table S2 (provided in the revised supplement) and Section 3.1, we performed transcriptome-wide screening for oxaliplatin resistance-associated genes. To prioritize candidates for functional validation: 1. Selection criteria: Genes with |logâ‚‚FC| > 4 (adjusted p < 0.01) were identified as high-confidence hits. 2. Functional screening: Nine top candidate genes meeting this threshold were selected. We constructed gene-specific knockdown plasmids for each and transfected them into HCT116 colon cancer cells. 3. Validation assay: Transfected cells were treated with oxaliplatin (48 hours), followed by quantification of surviving cells. 4. Key finding: As shown in Figure S2, DIRAS1 knockdown significantly potentiated oxaliplatin sensitivity (p < 0.05).
Comments 3: DIRAS1 Upregulation Mechanism:
Although it may not be the primary focus of this study, it would strengthen the manuscript if the authors could briefly discuss the underlying mechanisms responsible for DIRAS1 upregulation in CRC.
Response 3: Thank you for pointing this out. We agree with this comment. We sincerely appreciate the reviewer's valuable suggestion to explore the mechanistic underpinnings of DIRAS1 upregulation in colorectal cancer. This is elaborated in our discussion.
Comments 4: Knockdown and Rescue Experiments (Figure 2):
Key experiments should be repeated using multiple independent shRNAs or sgRNAs to validate the observed phenotypes. In addition, rescue experiments are essential to confirm the specific role of DIRAS1 in promoting cell proliferation.
Comments 7: Regulation of PHB1 by DIRAS1:
The mechanism by which DIRAS1 regulates PHB1 expression needs to be clearly described. Also, did the authors observe this correlation in larger patient cohorts or publicly available datasets?
Response 4/7: Thank you for pointing this out. We agree with this comment. Thank you for pointing this out. We agree with this comment. We sincerely appreciate the reviewer's insightful suggestion to delineate the DIRAS1-PHB1 regulatory hierarchy through bidirectional perturbation studies. To address this, we conducted two critical experiments: First, modulating PHB1 expression via overexpression and knockdown revealed no significant alteration in DIRAS1 protein levels, whereas DIRAS1 manipulation consistently altered PHB1 as previously reported. This unidirectional regulation was further validated in apoptosis assays using combinatorial approaches: DIRAS1 overexpression partially reversed the pro-apoptotic effect of PHB1 knockdown. Conversely, dual DIRAS1/PHB1 knockdown synergistically enhanced apoptosis, while dual overexpression showed no additive effect. These data robustly confirm PHB1 as a downstream executor of DIRAS1-mediated mitochondrial apoptosis, with no evidence of parallel pathway engagement. The complete datasets are presented in Supplementary Figure S7, and this mechanistic interpretation has been incorporated into the revised Results(Section 3.5) and Discussion.
Comments 5: Figure Legends and Scale Bars:
All figure legends lack details regarding the number of replicates performed. Additionally, scale bars are missing in Figure 6 and should be included for proper interpretation of the images.
Response 5: Thank you for pointing this out. We agree with this comment. Thank you for pointing this out. We agree with this comment. As suggested, we have completely revised all figure legend details to enhance clarity and completeness. Specifically, we have: 1. Added detailed experimental methods and scale bars. 2. Clearly defined experimental groups and the number of replicates performed. 3. Specified analysis conditions. 4. Included explicit data interpretations to guide readers through key findings. 5. Incorporated statistical methods.
Comments 6: DIRAS1 Induction by Oxaliplatin (Figure 3):
The manuscript claims that oxaliplatin treatment induces DIRAS1 expression, but the mechanism is unclear. This finding should be validated in additional CRC cell lines to confirm its generalizability.
Response 6: Thank you for pointing this out. We agree with this comment. We gratefully acknowledge the reviewer's insightful suggestion to validate the oxaliplatin-induced DIRAS1 upregulation across broader cellular contexts. To address this, we conducted extended time-course experiments in two additional CRC cell lines: Notably, DLD-1 cells (basally DIRAS1-null) exhibited no detectable DIRAS1 induction even after 72-hour oxaliplatin exposure, whereas SW620 cells demonstrated dose-dependent DIRAS1 elevation mirroring our HCT116 observations. This cell-type-specific response pattern—now documented in Supplementary Figure S8—suggests DIRAS1 inducibility may depend on pre-existing molecular contexts. While comprehensive mechanistic exploration across all cell lines was precluded by experimental constraints, we strategically prioritized HCT116 for deep mechanistic analysis given its: 1) representative DIRAS1 induction kinetics; 2) well-established use in chemoresistance studies.
Comments 8:Mitochondrial Damage Assessment:
The current data on mitochondrial damage requires more robust immunofluorescence evidence. Please include appropriate controls and additional experimental details to validate the findings.
Response 8: Thank you for pointing this out. We agree with this comment.We gratefully acknowledge the reviewer's insightful suggestions for enhancing methodological transparency and data interpretation. In response, we have implemented comprehensive revisions across both textual explanations and visual presentations. The figure 6 have been substantially upgraded through the addition of quantitative fluorescence intensity analyses, where statistical comparisons of mean fluorescence values (presented as mean ± SEM) now accompany original imaging data, with significance markers explicitly denoting key inter-group differences (p < 0.05, p < 0.01). Concurrently, we have meticulously re-evaluated and expanded the results interpretation sections to clarify the biological significance of experimental groupings—particularly distinguishing between DIRAS1 knockdown versus scramble control cohorts—while elaborating on the physiological relevance of critical metrics such as mitochondrial membrane potential (ΔΨm). Furthermore, every figure legend has been systematically enhanced to include: (i) detailed experimental protocols; (ii) unambiguous group definitions with sample sizes; (iii) specific analytical conditions; (iv) concise data interpretations contextualizing visual outputs; and (v) explicit descriptions of statistical methodologies applied, including normality testing, specific tests employed , and significance thresholds. These collective refinements substantially elevate the manuscript's reproducibility and analytical rigor.
Thank you again for your constructive comments, which have strengthened our work.
Sincerely,
Min Long

Round 2
Reviewer 1 Report
Comments and Suggestions for Authors
Thank you for your detailed responses to my suggestions and addressing my concerns. I believe the authors have improved the manuscript sufficiently for publication.
Reviewer 4 Report
Comments and Suggestions for Authors
Dear Editor:
The quality of this manuscript has been greatly improved. It is recommended to accept in the present form.